# A nascent riboswitch helix orchestrates robust transcriptional regulation through signal integration

Adrien Chauvier [1], Shiba S. Dandpat [1,2], Rosa Romero[1] & Nils G. Walter [1] ✉

Widespread manganese-sensing transcriptional riboswitches effect the dependable gene regulation needed for bacterial manganese homeostasis in changing environments. Riboswitches – like most structured RNAs – are believed to fold co-transcriptionally, subject to both ligand binding and transcription events; yet how these processes are orchestrated for robust regulation is poorly understood. Through a combination of single-molecule and bulk approaches, we discover how a single $Mn^{2+}$ ion and the transcribing RNA polymerase (RNAP), paused immediately downstream by a DNA template sequence, are coordinated by the bridging switch helix P1.1 in the representative *Lactococcus lactis* riboswitch. This coordination achieves a heretofore-overlooked semi-docked global conformation of the nascent RNA, P1.1 base pair stabilization, transcription factor NusA ejection, and RNAP pause extension, thereby enforcing transcription readthrough. Our work demonstrates how a central, adaptable RNA helix functions analogous to a molecular fulcrum of a first-class lever system to integrate disparate signals for finely balanced gene expression control.

In order to thrive, bacteria must constantly adjust their gene expression to their ever-changing environment. Because of the competition between species, it is crucial for their survival that bacteria adapt quickly to transient nutritional resources as well as external threats such as antibiotics and toxins[1].

Modulation of gene expression can be achieved in multiple ways. One mechanism bacteria employ is based on riboswitches, structural non-coding RNA elements mostly found in the 5′ untranslated regions (5′ UTR) of messenger RNA (mRNA)[2,3]. Bacterial riboswitches exquisitely determine the outcome of gene expression either at the level of transcription elongation or translation initiation[4]. A typical riboswitch contains two interwoven domains: (1) a conserved aptamer region able to recognize a specific ligand whose binding induces conformational changes and (2) a less conserved expression platform that modulates the expression level of the downstream gene(s) upon signal transduction from the aptamer[5]. In the case of transcriptional regulation, ligand binding to the aptamer modulates the formation or disruption

of a terminator hairpin in the expression platform that leads to the premature termination of transcription[6]. In addition, it has been shown that structural alterations of the nascent RNA as a function of ligand binding can drive the recruitment of Rho termination factor to the nascent mRNA to promote transcription termination[7–9].

The *yybP-ykoY* RNA motif, known to sense manganese ions ($Mn^{2+}$), is a particularly widespread class of bacterial riboswitches that has been found in many human and plant pathogens, and regulates genes associated with $Mn^{2+}$ homeostasis through transcription or translation control[10,11]. It does so by selectively sensing $Mn^{2+}$ at sub-millimolar levels over the predominant, millimolar magnesium ion ($Mg^{2+}$) and any other divalents found in the cell[12–14]. Crystal structures of the riboswitch aptamer domain revealed a pair of helices P1 and P2 stacking on top of each other, as well as the pair of P3 and P4, together forming a four-way junction (4WJ) that sandwiches two binding pockets formed by loops L1 and L3 for selective sensing of each one $Mg^{2+}$ and $Mn^{2+}$ ion, respectively (Fig. 1a)[12,13]. Single-molecule Förster Resonance Energy

[1]Single Molecule Analysis Group and Center for RNA Biomedicine, Department of Chemistry, University of Michigan, Ann Arbor, MI, USA. [2]Present address: Intel Corporation, Hillsboro, OR, USA. ✉e-mail: nwalter@umich.edu

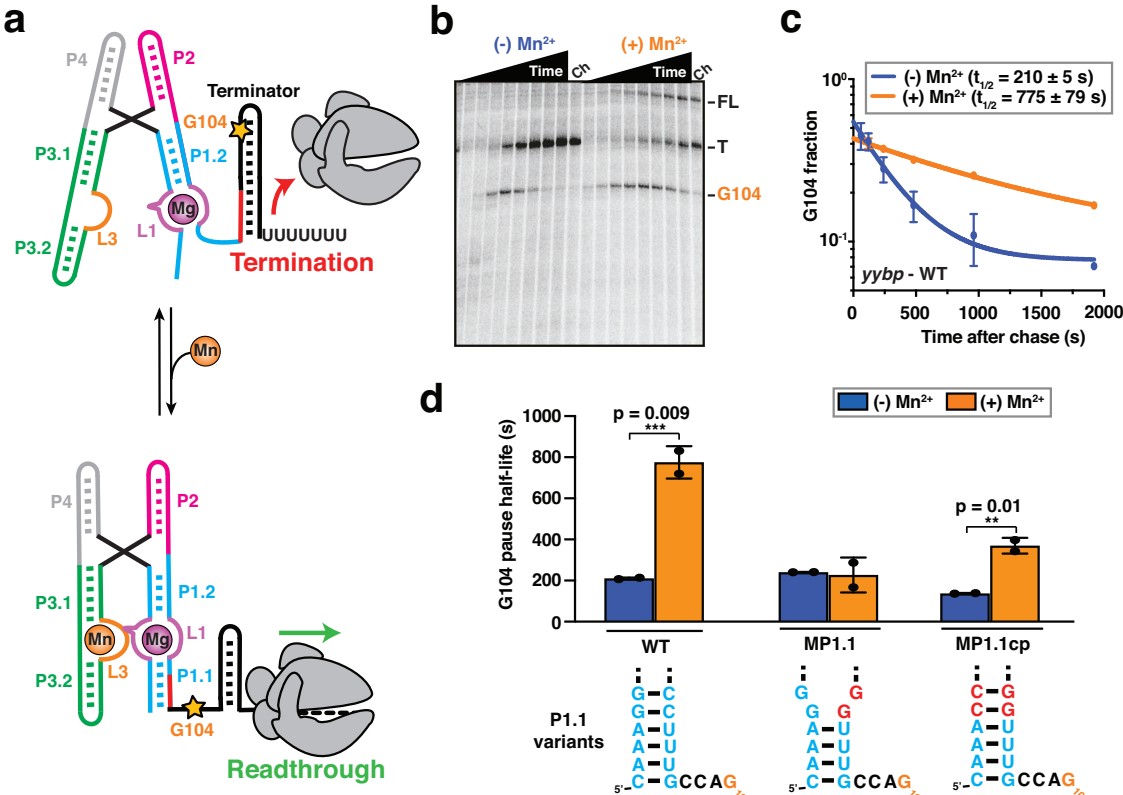

**Fig. 1 | Folding of P1.1 helix upon $Mn^{2+}$ binding stabilizes transcriptional pausing. a** The $Mn^{2+}$-sensing riboswitch from *L. lactis* regulates gene expression at the transcriptional level. Stabilization of the P1.1 helix upon $Mn^{2+}$ binding (Mn) prevents the formation of an intrinsic terminator hairpin, leading of the transcription of the downstream gene. Position of the G104 pause site is indicated as an orange star on the secondary RNA structure. **b** Representative denaturing gel showing the RNAP pauses during the transcription of the $Mn^{2+}$ riboswitch. Position of the pause (G104), termination (T) and full-length (FL) products are indicated on the right. Experiments were performed using 25 $\mu$M rNTPs in the absence ($-Mn^{2+}$) and presence ($+Mn^{2+}$) of 0.5 mM $Mn^{2+}$. The chase lanes (Ch) were taken at the end of the time course after an additional incubation with 500 $\mu$M rNTPs for 5 min. Unprocessed images are provided in the Supplementary Information. **c** Fraction of complexes at the G104 pause as a function of the transcription time in the absence (blue) and presence (orange) of 0.5 mM $Mn^{2+}$. The reported errors are the SD of the mean from $n = 2$ independent replicates. **d** Quantification of the G104 pause half-life in the WT, MP1.1 and MP1.1cp riboswitch variants. Experiments were performed using 25 $\mu$M rNTPs in the absence (blue) and presence (orange) of 0.5 mM $Mn^{2+}$. Error bars are the SD of the mean from $n = 2$ independent replicates. The statistical significance of differences was determined using the two-tailed Student's $t$-test ($***p < 0.01$, $**p < 0.05$, $*p < 0.1$). The predicted conformation of the P1.1 is represented below each riboswitch variant dataset. Source data are provided with this paper at https://doi.org/10.7302/22513.

Transfer (smFRET) studies further revealed that, while the presence of a physiological concentration of $Mg^{2+}$ pre-organizes the riboswitch into a dynamically docked global structure, capture of $Mn^{2+}$ stabilizes the docked conformation further to up-regulate transcription via antiterminator stabilization[13,15]. Formation of the P1.1 "switch" helix – distal from the $Mn^{2+}$ binding site, but proximal to the RNA polymerase (RNAP) – is critical for antitermination since in the absence of cognate ligand its 3' segment partitions toward terminator hairpin formation (Fig. 1a). As with most riboswitches, it is known that the aptamer plays a critical role in initiating the downstream transcription regulation events, however, the biological interplay of co-transcriptional RNA folding, tertiary structure dynamics, switch helix formation, and proximal RNAP is still poorly understood[16].

For regulatory RNA elements such as riboswitches, co-transcriptional folding events may tightly control the outcome of gene expression, by placing ligand binding and the speed of functional RNA structure adoption in competition with the rate of transcription[7,17]. This competition is expected to lead to kinetic control of gene regulation rather than the thermodynamic control invoked by the global free energy landscape of RNA folding[18,19], rendering the true biological context challenging to capture. Recent studies of riboswitches in the context of a paused elongation complex (PEC), i.e., in the presence of the DNA template and paused RNAP, have revealed that conformational changes of riboswitches are heavily influenced by the

transcription machinery[20–25], demonstrating the importance of the proximal RNAP for a mechanistic understanding of riboswitch-mediated gene regulation. While determinant, the kinetics of co-transcriptional riboswitch folding are difficult to observe at the short timescale dictated by transcription elongation[16,26].

Intrinsic RNAP pausing is an off-pathway kinetic step that interrupts the nucleotide addition cycle and can exert an additional layer of transcriptional control[27]. RNAP pause sites have been shown to play important roles in modulating gene expression, including transcription-translation coupling[28], RNA folding[29,30], transcription termination[6,31] and transcription factor recruitment[32,33]. RNAP pausing relies on specific DNA sequences embedded throughout the genome[34], and can be further promoted by the formation of RNA structures in the vicinity of the RNA exit channel[35] and/or the interaction with specific transcription factors[36].

N-utilizing substance A protein (NusA) is an essential transcription factor that modulates transcription speed using a multitude of protein and RNA binding interactions. It is required along with other transcription factors for the processive transcription of ribosomal RNA[37,38] and is usurped as a component of the phage lambda protein N-mediated antitermination system[39,40]. NusA can promote transcription termination[31,41,42], decrease the overall speed of RNAP[23,43], stimulate hairpin-stabilized pauses[44–46], and participate in the correct folding of the RNA component of RNase P[30,47]. NusA is a multi-domain protein

whose N-terminal domain is necessary and sufficient for stimulating RNAP pausing by directly contacting the RNAP exit channel[44,45]. This domain is followed by three RNA binding domains S1, KH1 and KH2[44,45,48] and, finally, in *Escherichia coli* (*E. coli*), the NusA C-terminal domain with two acidic repeats, AR1 and AR2, that directly interact with the alpha subunit of RNAP[49]. The latter contact induces release of the auto-inhibitory AR2-KH1 interaction, which then allows the S1, KH1 and KH2 domains to also interact with the nascent RNA transcript[50]. Despite this detailed structural understanding and the proximity of NusA and the nascent RNA transcript at the RNAP exit tunnel, to date, there are only few examples for NusA modulating riboswitch-based transcription regulation[17,23,51,52]. Specifically, it was demonstrated that the time window for ligand sensing is widened in the presence of NusA, therefore allowing for a more efficient gene regulatory response[17,51,52]. However, a recent study has uncovered a dynamic competition between NusA and ligand binding to a fluoride-sensing riboswitch, highlighting an active interplay between transcription factor recruitment and co-transcriptional RNA folding within a PEC[23].

In the present work, we used the well-studied, representative Mn$^{2+}$-sensing riboswitch from *Lactococcus lactis* (*L. lactis*) as a model system to investigate the impact of the transcription machinery on a highly structured nascent RNA. By employing a combination of single-molecule and biochemical approaches to a reconstituted PEC, we discovered an intricately balanced mechanism wherein the presence of both RNAP and Mn$^{2+}$ assists tertiary structure docking of the riboswitch and thereby actively contributes to the final gene regulatory outcome through co-transcriptional antiterminator folding. Examining the P1.1 switch helix revealed that it is stabilized by sensing the distal, Mn$^{2+}$-dependent docking as well as the presence of downstream RNAP when a DNA template sequence site-specifically pauses it in proximity. We further uncovered a role for transcription factor NusA in favoring P1.1 formation, independent of and redundant, but also competing, with Mn$^{2+}$ action. Overall, our results provide a comprehensive mechanism for how a multitude of distinct intracellular signals influence co-transcriptional folding of a riboswitch by their integration through the central P1.1 switch helix, which acts like the fulcrum of a first-class lever system to adapt bacterial gene expression dependably to an ever-changing environment.

## Results

### Ligand-induced stabilization of riboswitch helix P1.1 modulates RNAP pausing efficiency

Transcriptional pausing has been shown to be an inherent parameter that enables co-transcriptional RNA folding and protein recruitment to play an active role in regulating gene expression[27,35]. Therefore, we evaluated how riboswitch folding modulates transcriptional pausing, using single-round transcription assays (Fig. 1b). We identified a long-lived intrinsic (elemental) pause site at position G104 in the *yybP* expression platform, which is four nucleotides downstream of the P1.1 switch helix (Fig. 1b and Supplementary Fig. 1) and closely related to the known bacterial consensus pause sequence in the DNA template[34]. Interestingly, the duration of the G104 pause was significantly extended in the presence of saturating Mn$^{2+}$ ions (Fig. 1c and Supplementary Table 2), indicating that the ligand-bound state further promotes RNAP pausing at this position. Control experiments performed with a mutant abolishing Mn$^{2+}$ binding to the riboswitch (A41U)[13] showed no such difference, supporting that promotion of RNAP pausing is the result of riboswitch folding in the ligand-bound state, rather than a direct effect on the enzyme (Supplementary Table 2).

Upon examination of the riboswitch secondary structure relative to the G104 pause position in the RNAP active site, we hypothesized that the formation of the P1.1 switch helix inside the RNAP exit channel stabilizes transcriptional pausing (Supplementary Fig. 1). To test its involvement in RNAP pausing, we destabilized the P1.1 helix by

mutating residues C95 and C96 to G95 and G96, respectively (mutant MP1.1; Fig. 1d and Supplementary Table 2). Consistent with our hypothesis, MP1.1 completely abolishes the impact of Mn$^{2+}$ on RNAP pausing efficiency. Importantly, compensatory second-site mutations G5C and G6C (mutant MP1.1cp) not only restore two base pairs at the top of P1.1 but also the Mn$^{2+}$-induced enhancement of RNAP pausing to levels similar to those of the wild-type (Fig. 1d and Supplementary Table 2).

Overall, these data are consistent with a regulatory system in which the ligand-mediated stabilization of RNA helix P1.1 in the RNAP exit channel extends transcriptional pausing.

### Ligand-mediated folding of the P1.1 helix depends on the proximity of RNAP

To characterize the co-transcriptional folding of the riboswitch P1.1 helix as a function of ligand binding, we used an in vitro transcription assay that generates a co-transcriptionally folded PEC with a single fluorophore at the 5' end for identification and subsequent probing at the single-molecule level (Fig. 2a, b)[22]. We added a short (8-nucleotide), Cy5-labeled DNA SiM-KARTS (Single-Molecule Kinetic Analysis of RNA Transient Structure) probe against the 5' segment of the P1.1 helix, which allowed for dynamic probing of solvent accessibility of the C2-G8 switching region (Fig. 2b and Supplementary Fig. 1)[23,53,54]. Upon simultaneous excitation of Cy3 and Cy5, single-molecules were located on the microscope slide through their Cy3 (green) PEC-104 fluorescence signal, whereas the accessibility of the target switching region over time was detected through spikes of the corresponding SiM-KARTS fluorescence intensity in the Cy5 (red) channel (Fig. 2c, d). The cumulative dwell time distributions of the SiM-KARTS probe in the 5' P1.1 unbound ($\tau_{unbound}$) and bound ($\tau_{bound}$) states were fitted with double-exponential functions to extract the corresponding binding ($k_{SK\text{-}on}$) and dissociation ($k_{SK\text{-}off}$) rate constants, respectively (Fig. 2e, f).

In the presence of near-physiological 1 mM Mg$^{2+}$ only, the SiM-KARTS probe bound to its 5' P1.1 target with two rate constants of $3.67 \pm 0.02 \times 10^6$ M$^{-1}$ s$^{-1}$ ($k_{SK\text{-}on,fast}$; fraction 59%) and $0.63 \pm 0.01 \times 10^6$ M$^{-1}$ s$^{-1}$ ($k_{SK\text{-}on,slow}$; fraction 41%), as well as dissociated with two rate constants of $0.55 \pm 0.01$ s$^{-1}$ ($k_{SK\text{-}off,fast}$; fraction 61%) and $0.08 \pm 0.01$ s$^{-1}$ ($k_{SK\text{-}off,slow}$; fraction 39%). This observation is consistent with the notion that the SiM-KARTS probe senses (at least) two alternative structures of the riboswitch P1.1 helix that could arise on the same RNA molecule. Indeed, the targeted C2-G8 region is expected to either form the P1.1 helix or adopt a single-stranded conformation, which the absence of Mn$^{2+}$ ligand should favor[12,13].

To test the latter notion, we next transcribed PEC-104 in the presence of a saturating concentration (0.5 mM) of Mn$^{2+}$ (generally added together with 0.5 mM Mg$^{2+}$ for a consistent total concentration of 1 mM divalents) and again probed the 5' P1.1 segment using SiM-KARTS (Fig. 2d). Under these conditions, the probe binding rate constants were $2.93 \pm 0.01 \times 10^6$ M$^{-1}$ s$^{-1}$ ($k_{SK\text{-}on,fast}$; fraction 44%) and $0.49 \pm 0.01 \times 10^6$ M$^{-1}$ s$^{-1}$ ($k_{SK\text{-}on,slow}$; fraction 56%), whereas the two dissociation rate constants were $0.36 \pm 0.01$ s$^{-1}$ ($k_{SK\text{-}off,fast}$; fraction 85%) and $0.08 \pm 0.01$ s$^{-1}$ ($k_{SK\text{-}off,slow}$; fraction 15%). Notably, both the faster and slower binding rate constants decreased by ~20% compared to the no-ligand condition (Fig. 2e). In addition, the fractional contribution of $k_{SK\text{-}on,slow}$ increased (from 41% to 56%) when co-transcriptionally adding Mn$^{2+}$, at the expense of the $k_{SK\text{-}on,fast}$ fraction. Consequently, the weighted average binding rate constant ($k_{SK\text{-}on}^{overall}$) decreased by ~37% (from $2.42 \pm 0.17 \times 10^6$ M$^{-1}$ s$^{-1}$ to $1.53 \pm 0.15 \times 10^6$ M$^{-1}$ s$^{-1}$) upon addition of Mn$^{2+}$ ions (Fig. 2e and Supplementary Table 3). These observations support the notion that ligand binding to the riboswitch in the context of PEC-104 promotes P1.1 helix formation and thus limits SiM-KARTS probe access, albeit not completely. Notably, binding of the SiM-KARTS probe effectively mimics formation of helix P1.1, albeit as an RNA:DNA hybrid (Fig. 2b). The observed slight decrease in overall SiM-

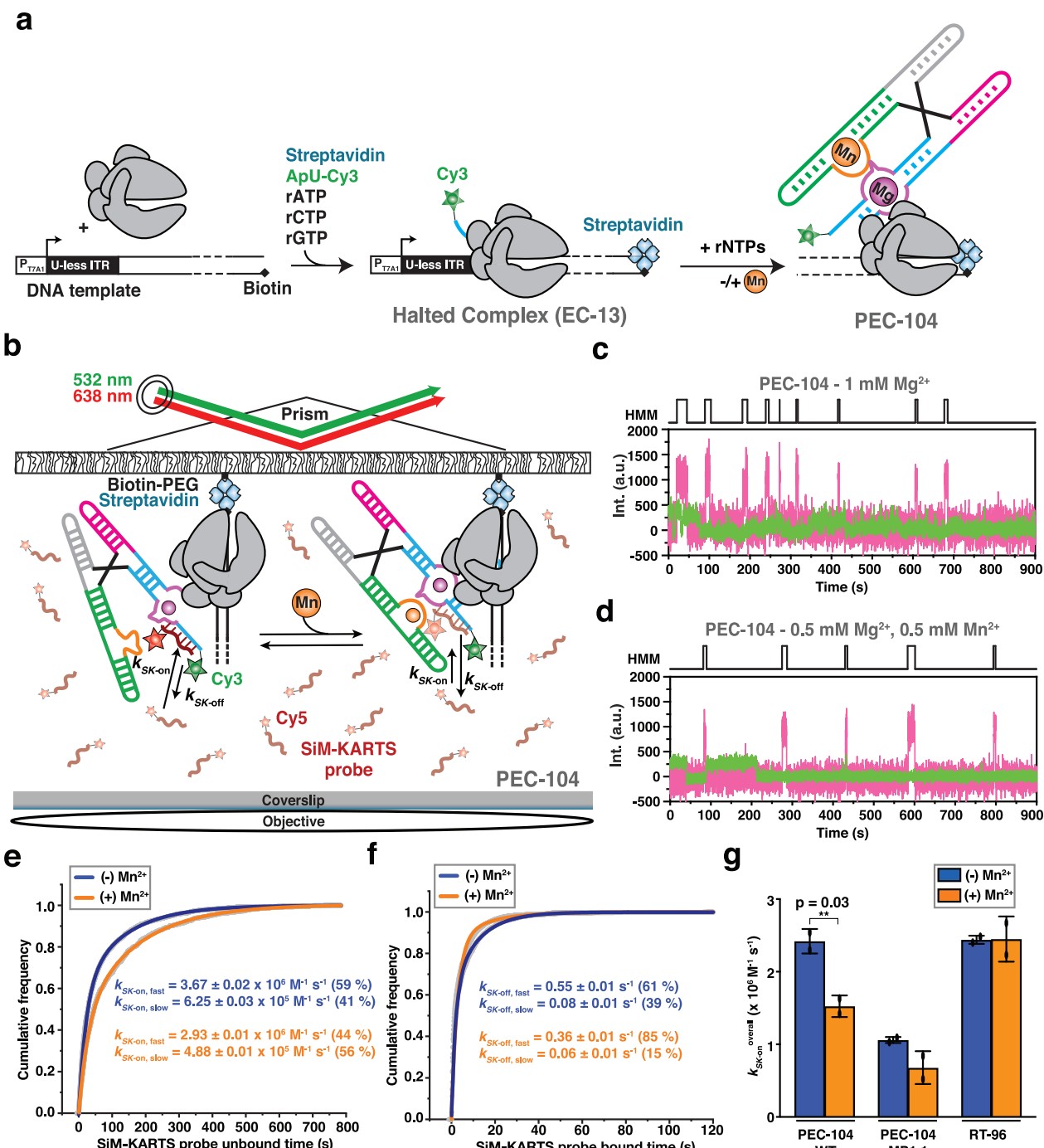

**Fig. 2 | Probing P1.1 structural dynamics of co-transcriptionally folded transcripts. a** Nascent fluorescently labeled Paused Elongation Complexes (PEC) are transcribed in vitro using *E. coli* RNAP. The halted complex (EC-13) is prepared through the addition of a dinucleotide labeled with Cy3 (ApU-Cy3) and UTP deprivation (ATP/CTP/GTP) to halt the RNAP at the end of the U-less ITR. The biotin-streptavidin interaction at the 3′-end of the DNA template constitutes a stable transcriptional roadblock to stall the RNAP at the desired position. **b** SiM-KARTS experimental setup. Immobilization of the PEC on the microscope slide occurs through the biotin-streptavidin roadblock. Repeated bindings of the SiM-KARTS probe targeting the 5′ segment of P1.1 are monitored through direct excitation of the Cy5 fluorescent dye. Representative single-molecule trajectories showing the SiM-KARTS probe binding (pink) to 5′ segment of the P1.1 helix of PEC-104 recorded in the absence (**c**) or presence (**d**) of 0.5 mM $Mn^{2+}$ added cotranscriptionally. Hidden Markov modeling (HMM) is indicated on the top of each trace. A.U., arbitrary unit. Plots displaying the cumulative unbound (**e**) and bound

(**f**) dwell times of the SiM-KARTS probe in the absence (blue) and presence (orange) of 0.5 mM $Mn^{2+}$ in the context of PEC-104. The binding ($k_{on}$) and dissociation ($k_{off}$) rate constants of the SiM-KARTS probe are indicated. The reported errors are the error of the fit. Total number of molecules analyzed for each condition is as follow: (−) $Mn^{2+}$ = 350; (+) $Mn^{2+}$ = 246. **g** Overall binding rate constants ($k_{on}$) of the SiM-KARTS probe in the context of PEC-104 WT, MP1.1 and RT-96 constructs determined in the absence (blue) and presence (orange) of 0.5 mM $Mn^{2+}$. Error bars are the SD of the mean from independent replicates. The statistical significance of differences was determined using the two-tailed Student's *t*-test (***$p < 0.01$, **$p < 0.05$, *$p < 0.1$). See also Supplementary Fig. 2. Total number of molecules analyzed for each condition is as follow: PEC-104 WT (−) $Mn^{2+}$ = 350; PEC-104 WT (+) $Mn^{2+}$ = 246; PEC-104 MP1.1 (−) $Mn^{2+}$ = 330; PEC-104 MP1.1 (+) $Mn^{2+}$ = 292; RT-96 (−) $Mn^{2+}$ = 118; RT-96 (+) $Mn^{2+}$ = 182. Source data are provided with this paper at https://doi.org/10.7302/22513.

KARTS probe dissociation rate constant $k_{SK\text{-off}}^{overall}$ (from 0.37 s$^{-1}$ to 0.32 s$^{-1}$; Fig. 2f and Supplementary Table 3) upon addition of Mn$^{2+}$ is therefore also consistent with the ligand stabilizing formation of the P1.1 helix. As a control experiment, we performed SiM-KARTS probing of a PEC-104 with our mutant MP1.1 in which nucleotides C95 and C96 were both mutated to G, disrupting the upstream Watson-Crick base pairs of the P1.1 helix (Supplementary Fig. 1). While the overall SiM-KARTS binding rate constant expectedly decreased, there was still a similar decrease (35%) associated with the addition of Mn$^{2+}$ (Fig. 2g, Supplementary Fig. 2 and Supplementary Table 3), suggesting that the ligand influences the entire P1.1 helix surveyed by the SiM-KARTS probe.

To ask whether the proximal RNAP of PEC-104 participates in the ligand-dependent modulation of the P1.1 helix, we next performed SiM-KARTS probing on a co-transcriptionally folded transcript from which the RNAP was run off by omission of the 3' biotin-streptavidin block. To this end, transcription was performed on an extended DNA template, installing an additional 3' end RNA sequence as "anchor" that was hybridized with a 5' biotinylated capture probe (CP) for subsequent immobilization of the transcript on the microscope slide[23]. Importantly, this released transcript (RT-96) comprised the full *yybp* riboswitch up to residue C96 to allow for formation of the upper base pairs of the P1.1 helix, mimicking the entire RNA transcript upstream of the RNA-DNA hybrid in the PEC. Strikingly, no effect of Mn$^{2+}$ on the RT-96 riboswitch was detected by the SiM-KARTS probe, yielding comparable rate constants in the presence and absence of ligand that match the kinetic parameters for PEC-104 in the absence of ligand, where the P1.1 helix is not (fully) formed (Fig. 2g, Supplementary Fig. 2 and Supplementary Table 3). These observations indicate that the presence of RNAP in PEC-104 actively assists P1.1 helix folding upon addition of Mn$^{2+}$.

Taken together, our single-molecule probing reveals that ligand and RNAP, paused at a specific downstream site, collaborate to nucleate the P1.1 switch helix. Put differently, the riboswitch integrates disparate signals from distal Mn$^{2+}$ binding and proximal RNAP pausing for gene regulatory action.

### The presence of RNAP promotes the docked conformation of the riboswitch

Our biochemical and single-molecule probing assays underscore the importance of both RNAP and ligand for riboswitch folding. However, SiM-KARTS only reveals structural rearrangements that occur locally (P1.1 stem) and does not report the intramolecular dynamics for a single RNA molecule. Generally, the presence of RNAP and DNA template during transcription has been shown to affect global folding of structural RNAs, such as riboswitches, which in turn modulates the output of gene expression[7,20–23,28]. Therefore, we next evaluated how the proximal RNAP modulates the global docking dynamics when paused at G104 (PEC-104), using smFRET between fluorophores attached to P1 and P3 of a hybridized two-strand riboswitch (Fig. 3a and Supplementary Fig. 1), as previously established for the isolated aptamer with stabilized P1.1[13,15].

At near-physiological concentration of Mg$^{2+}$ (1 mM), the riboswitch in PEC-104 sampled two predominant conformational states in which the low- (-0.2) and high-FRET (-0.7) populations represent the undocked and docked states, respectively (Fig. 3b, d), consistent with previous studies of the isolated aptamer[13,15]. The FRET histogram showed nearly equal populations in the low- and high-FRET regimes (46% and 53%, respectively), indicating that the riboswitch can occupy the docked state significantly in the absence of Mn$^{2+}$. Upon addition of 0.5 mM Mn$^{2+}$, the riboswitch adopted a more stably docked conformation (Fig. 3c) with the high-FRET population increasing to -70%, indicating that binding of Mn$^{2+}$ shifts the equilibrium toward the docked state (Fig. 3d and Supplementary Table 4), as previously observed absent the PEC[13,15].

To assess the influence of RNAP on the global dynamics of (un)docking, we assembled the same FRET-labeled riboswitch and immobilized it to the microscope slide through a biotinylated locked nucleic acid (LNA) capture probe using the same base pairing employed to reconstitute PEC-104 (LNA-104; Fig. 3a). As illustrated by the presence of the two predominant low- (-0.2) and high-FRET (-0.7) populations, the riboswitch in the absence of RNAP adopted the same two dynamic conformational states as in PEC-104 (Fig. 3e). Notably, in the presence of Mg$^{2+}$ ions only, the docked population without RNAP was significantly less populated, with 42% for LNA-104 compared to 53% under the same conditions for PEC-104 (Fig. 3d, e), suggesting that the proximal RNAP promotes the formation of the docked state. Similarly, addition of 0.5 mM Mn$^{2+}$ led to 65% docked state for LNA-104, but a slightly increased 67% for PEC-104 (Fig. 3d, e); whereas in 100 mM K$^+$ without divalents the riboswitch adopted only 24% docked state for LNA-104, which rose to 35% for PEC-104 (Supplementary Fig. 3 and Supplementary Table 4). These observations demonstrate that RNAP favors docking under a broad set of conditions.

As control, we designed an RNA that went ten bases beyond the G104 pause, and immobilized it through a complementary LNA capture strand (LNA-114) to mimic the structure after full transcription of the riboswitch with RNAP far downstream (Supplementary Fig. 10). In the presence of Mg$^{2+}$ only, just 25% of the LNA-114 riboswitch were found in the high-FRET (-0.7) docked state, compared to 42% for LNA-104 and 53% for PEC-104 (Supplementary Table 4), suggesting that extension beyond the G104 pause disfavors docking. As expected, with the addition of Mn$^{2+}$ the fraction of docked state significantly increased to 67 % (Supplementary Fig. 10 and Supplementary Table 4), indicating that the full-length riboswitch remains highly Mn$^{2+}$ sensitive, even as RNAP has already translocated further downstream.

Taken together, our analysis so far reveals that the presence of RNAP at the G104 pause site promotes the docked riboswitch conformation, particularly in the absence of ligand, due to dynamic sampling of a partially folded P1.1 switch helix.

### A semi-docked riboswitch conformation represents the initial nucleation of P1.1

From our previous work it is known that binding of Mn$^{2+}$ stabilizes a static docked (SD) conformation in the riboswitch in the presence of a complete P1.1 stem[13]. Further inspection of the smFRET trajectories in our 104 constructs revealed that HMM modeling was best performed using three FRET states (Undocked = U (-0.2), Semi-Docked = D* (-0.45), Docked = D (-0.75)). In the presence of Mg$^{2+}$ only, -70% of all riboswitch molecules in PEC-104 sampled these three FRET states dynamically over minor stable undocked (SU, -9%) and SD populations (-20%; Fig. 4a, b). Transitions from the U to the D state ($k_{dock}$) and from D to U ($k_{undock}$) occurred at comparable rate constants of -2.2 s$^{-1}$ and -2.4 s$^{-1}$, implying facile reversibility of docking. In the absence of RNAP in LNA-104, the rate constants shifted to -1.1 s$^{-1}$ ($k_{dock}$) and -5.5 s$^{-1}$ ($k_{undock}$; Fig. 4e, f), consistent with our previous conclusion that RNAP promotes docking. Likewise, transitions from U or D to D* and from D* to U or D were more prominent in PEC-104 than in LNA-104, suggesting that the semi-docked D* is also favored by the RNAP. A relatively slow $k_{dock,U\rightarrow D^*}$ transition of -1.8 s$^{-1}$ and a faster $k_{undock,D^*\rightarrow U}$ of -2.9 s$^{-1}$ in PEC-104, combined with an even faster $k_{dock,D^*\rightarrow D}$ of -3.03 s$^{-1}$, suggest that the D* state is a frequent, albeit non-obligatory, intermediate between the undocked and undocked states. These findings show that, in the presence of a near-physiological Mg$^{2+}$ concentration, the riboswitch within the PEC can sample the previously unobserved semi-docked state D*, in addition to the previously reported U and D states.

As expected from the FRET population histograms, addition of Mn$^{2+}$ increased the proportion of the on-diagonal SD population in both PEC-104 (from -20% to -35%; Fig. 4c, d) and LNA-104 (from -18% to -46%). As a result, the dynamic population transitioning between the

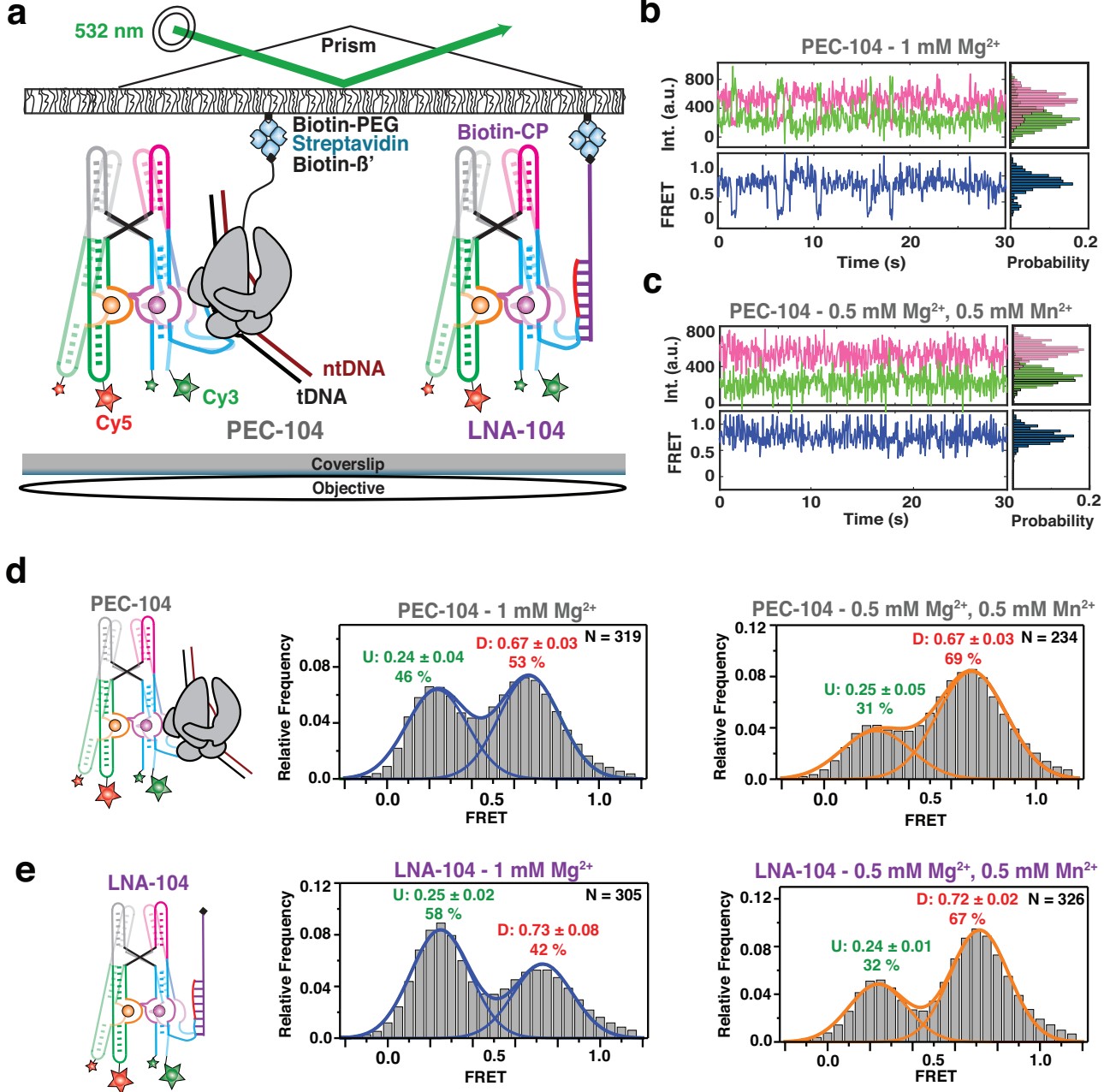

**Fig. 3 | Investigation of Mn²⁺ riboswitch docking in the presence and absence of RNAP by smFRET. a** smFRET experimental setup. Paused Elongation Complex (PEC) immobilization utilizes biotinylated *E. coli* RNAP. Locked Nucleic Acid (LNA) immobilization utilizes a complementary biotinylated LNA capture probe. Location of donor (Cy3 – green) and acceptor (Cy5 – red) fluorophores are indicated. Representative donor-acceptor (top) and smFRET (bottom) traces for PEC-104 in the absence (**b**) and presence (**c**) of 0.5 mM Mn²⁺. Probability FRET histogram for each trace is indicated on the right. smFRET histograms of PEC-104 (**d**), LNA-104 (**e**) in the presence of only Mg²⁺ (middle panel) and in the presence of both Mg²⁺ and Mn²⁺ (right panel). Cartoon of each construct is indicated on the left. The SD of each FRET value is reported. The total number of traces (N) included in each histogram is indicated on top right corner. Source data are provided with this paper at https://doi.org/10.7302/22513.

U, D* and D states decreased to ~58%, supporting that ligand binding shifts the equilibrium toward a more compact, stably docked conformation. Indeed, the transition rate constant from U to D slightly increased from 2.16 s⁻¹ in the absence to 2.29 s⁻¹ in the presence of Mn²⁺, while the transition from D to U became slower (2.41 s⁻¹ and 1.67 s⁻¹ in the absence and presence of Mn²⁺, respectively). Binding of Mn²⁺ significantly shifted the transition towards higher D* and D populations, with a smaller population now sampling the U state (Fig. 4d). Similarly, the U to D* transitions became faster in the presence of Mn²⁺ (2.12 s⁻¹ compared to 1.76 s⁻¹ with only Mg²⁺), whereas the transition from D to D* became slower (1.51 s⁻¹ compared to 1.96 s⁻¹

with only Mg²⁺), further supporting that the D* state represents a semi-docked state on route to Mn²⁺-induced full docking.

In the absence of RNAP (LNA-104), the riboswitch was less dynamic than in PEC-104, preferentially adopting the SU (~30%) and SD (~46%) populations in the absence and presence of Mn²⁺, respectively (Fig. 4e–h). In addition, $k_{dock}$ was slower and $k_{undock}$ faster than in the absence than the presence of RNAP (Fig. 4f, h), further supporting that proximal RNAP promotes the adoption of the D state. LNA-104 docking was closer to a two-state type transition than PEC-104, with the D* state somewhat less populated. This was particularly true in the presence of Mn²⁺, in which the D* to D transitions increased from 2.44 s⁻¹ in the

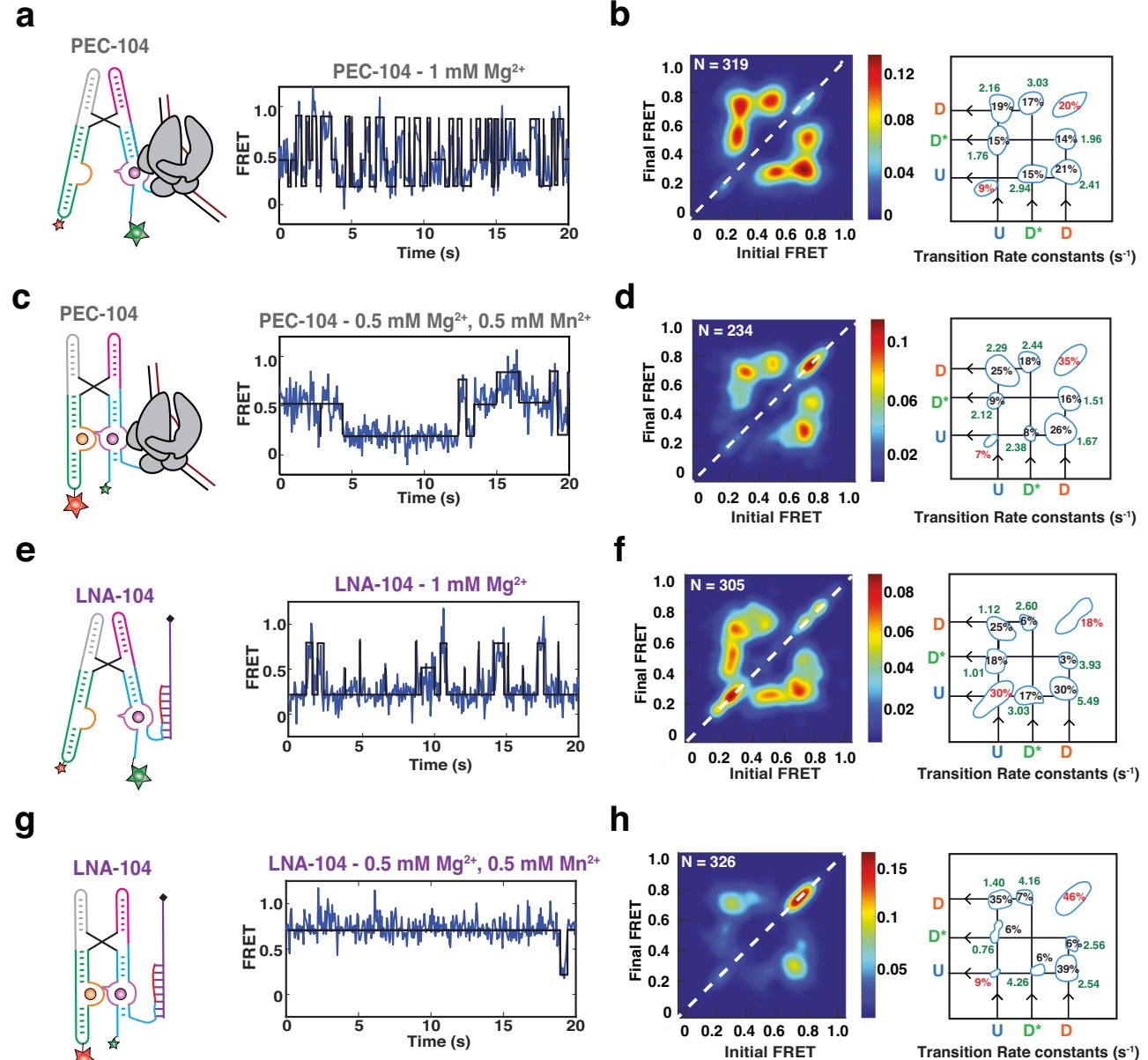

**Fig. 4 | Effect of RNAP and Mn²⁺ on the conformational dynamics of the riboswitch.** Representative FRET trajectories for PEC-104 in the presence of only Mg²⁺ (**a**) and in the presence of both Mg²⁺ and Mn²⁺ (**c**) along with the corresponding Transition Occupancy Density Plots (TODP) and the transition rate constants calculated in the absence (**b**) and in the presence (**d**) of Mn²⁺, See also Supplementary Figs. 5 and 6. **e**–**h** Representative FRET trajectories for LNA-104 in the presence of only Mg²⁺ (**e**) and in the presence of both Mg²⁺ and Mn²⁺ (**g**) along with the corresponding Transition Occupancy Density Plots (TODP) and the transition rate constants calculated in the absence (**f**) and in the presence (**h**) of Mn²⁺, See also Supplementary Figs. 8 and 9. TODPs represent dynamic traces as "off-diagonal" and the static traces as "on-diagonal" contour, where the color scale shows the prevalence of each population. The riboswitch conformational state corresponding to each FRET value is indicated. U = Undocked, D* = Pre-Docked, D = Docked. Source data are provided with this paper at https://doi.org/10.7302/22513.

presence to 4.16 s⁻¹ in the absence of RNAP. This observation suggests that the RNAP can stabilize the semi-docked state, possibly through electrostatic and steric interaction with the riboswitch as observed in the preQ₁-sensing riboswitch PEC from *B. subtilis*[21].

As a further control, we analyzed the FRET transition rate constants in the extended LNA-114 construct in the absence and presence of Mn²⁺ ion (Supplementary Fig. 11). With a now-fully available P1.1 switch helix, the riboswitch behaved similarly to LNA-104, exhibiting a significant proportion of SU (42%) and SD (40%) in both the absence and presence of Mn²⁺, respectively. In addition, the D* state was observed to be much less accessed than in both LNA-104 and PEC-104, consistent with the notion that the semi-docked conformational state D* may depend on an only partially folded P1.1. Even with three-state

HMM fits, the TODP predominantly exhibited two-state transitions between the U and D states (Supplementary Fig. 11). In fact, the overall behavior resembled that of a manganese-sensing riboswitch with a stabilized P1.1[13], with $k_{undock}$ becoming slower upon ligand binding. Similarly, $k_{undock,D^* \to U}$ in the presence of Mn²⁺ was significantly faster for LNA-114 (7.14 s⁻¹) than LNA-104 (4.26 s⁻¹), indicating a less stable D* state when the P1.1 is fully stabilized. This observation also explains why D* was not significantly detected in the earlier study featuring the stabilized P1.1[13].

To further test whether the observed conformational changes are due to stabilization of switching helix P1.1 in PEC-104, we directly surveyed P1.1 folding using smFRET. To measure dynamic distances along the P1.1 helical axis, we employed stepwise transcription to

fluorescently label the 5'end of the transcript with Cy3 and incorporate an azido-modified NTP at position 11 (Supplementary Fig. 1), which subsequently was labeled using click-chemistry with DBCO-Cy5[22,55]. To stall the RNAP at position 104 (generating PEC-104), the DNA template was immobilized on streptavidin-coated magnetic beads using 3'desthio-biotin, which allowed for subsequent bead release via a competing biotinylated-DNA oligonucleotide (Supplementary Fig. 12).

At near-physiological concentration of $Mg^{2+}$ only (1 mM), helix P1.1 in PEC-104 sampled two predominant conformational states of mid- (~0.40) and high-FRET (~0.65) values and nearly equal populations (58% and 42%, respectively; Supplementary Fig. 13b). Upon addition of 0.5 mM $Mn^{2+}$, P1.1's high-FRET state became more prominent (Supplementary Fig. 13c), suggesting that it is associated with the tertiary docked state that is similarly stabilized by $Mn^{2+}$, consistent with our previous observations on the other FRET vector (Fig. 3). Notably, the majority of traces were static in one or the other FRET state, populating on-diagonal contours in the TODPs (Supplementary Fig. 13d, e). This finding may indicate either very fast transitions beyond the time resolution of our smFRET acquisition, similar to the short-lived (3 ms) rare ("excited") state of the fluoride-sensing riboswitch from *B. cereus*[56], or long-lived steric clashes with the nearby RNAP surface.

Taken together, our single-molecule kinetic analyses reveal a previously unobserved semi-docked, pre-folded riboswitch conformation involving the initial nucleation of P1.1 and stabilization by both RNAP and ligand binding.

## Transcription factor NusA recruitment stabilizes P1.1 but is ejected upon ligand binding

Our single-molecule and bulk data suggest a mechanism wherein the initial nucleation of P1.1 extends intrinsic RNAP pausing, allowing time for co-transcriptional folding of the riboswitch into its ligand-sensing conformation for gene regulation. Transcription factor NusA is known to promote transcriptional pausing in the context of hairpin-stabilized Class I pauses through interaction with both the nascent transcript and the RNAP[44,57,58]. Because folding of P1.1 within the RNAP exit channel could act similarly to a Class I transcriptional pause, we next evaluated RNAP pausing efficiency in the presence of NusA (Fig. 5a). Indeed, in the presence of $Mg^{2+}$ only, we observed a ~4-fold increase of the G104 half-life upon addition of NusA (210 s in the absence of versus 906 s in the presence of NusA; Fig. 5b, c and Supplementary Table 2). In contrast, in the presence of $Mn^{2+}$ NusA no longer has any effect on the G104 pause half-life (859 s in the absence versus 820 s in the presence of NusA). These findings suggest that the ligand-bound riboswitch with its stabilized P1.1 helix supersedes NusA-mediated enhancement of G104 pausing efficiency, as also seen for the fluoride-sensing riboswitch[23], suggesting redundant action by $Mn^{2+}$ and NusA.

Because we identified the P1.1 switch helix as the key modulator of the G104 pause in the absence of the transcription factor (Fig. 1d), we next tested the hypothesis that it might also negotiate NusA's ability to affect transcriptional pausing. Indeed, while the MP1.1 mutant completely abolished the effect of $Mn^{2+}$ on pausing efficiency (Fig. 1d), the mutant allowed NusA to retain its ability to extend pausing independent of $Mn^{2+}$ (Fig. 5c). Complementarily, when the compensatory mutant MP1.1cp was used, the wild-type behavior was restored, i.e., NusA's large effect on pausing in $Mg^{2+}$ only was lost upon addition of $Mn^{2+}$ (~4-fold and only ~1.5-fold enhancement of G104 pause half-life by NusA without and with ligand, respectively; Fig. 5c and Supplementary Table 2). These data further support that $Mn^{2+}$ and NusA have a redundant function in P1.1 formation. Interestingly, addition of NusA during in vitro transcription of the riboswitch moderately decreased the concentration of $Mn^{2+}$ needed to modulate transcription termination and readthrough (Supplementary Fig. 14a, b), suggesting that NusA may not only alter the transcription rate and thus time window for ligand sensing, but also modestly sensitize the riboswitch to its

ligand[17,23,51]. In further support, an in vitro termination assay performed in either the absence or presence of NusA revealed a small, but significant decrease in transcription readthrough in the presence of both ligand and transcription factor, from 60% in the absence to 43% in the presence of NusA (Supplementary Fig. 14c).

To understand the mechanism by which NusA and $Mn^{2+}$ alternatively affect G104 pause behavior, we directly monitored the dynamics of NusA's interaction with PEC-104 through an established single-molecule colocalization assay (Fig. 5d–f)[23]. In the presence of $Mg^{2+}$ only, NusA bound with two rate constants of $2.10 \pm 0.01 \times 10^7$ $M^{-1}$ $s^{-1}$ ($k_{NusA\text{-}on,fast}$; fraction 25%) and $3.06 \pm 0.01 \times 10^6$ $M^{-1}$ $s^{-1}$ ($k_{NusA\text{-}on,slow}$; fraction 75%), as well as two dissociation rate constants of $0.75 \pm 0.02$ $s^{-1}$ ($k_{NusA\text{-}off,fast}$; fraction 91%) and $0.11 \pm 0.02$ $s^{-1}$ ($k_{NusA\text{-}off,slow}$; fraction 9%; Fig. 5g, h and Supplementary Table 5). Similar to the SiM-KARTS probe, these biphasic kinetics suggest that P1.1 exists in (at least) two conformational states that NusA senses. Interestingly, the binding kinetics resemble those determined for NusA binding to a canonical Class I pause PEC (*his*PEC)[23], further supporting that the G104 pause is stabilized through the formation of the P1.1 switch helix within the RNAP exit channel.

In the presence of $Mn^{2+}$, the cumulative unbound dwell times could only be fitted with a single-exponential $k_{NusA\text{-}on}$ of $3.35 \pm 0.01 \times 10^6$ $M^{-1}$ $s^{-1}$ (Fig. 5g), similar to the slower binding rate constant in $Mg^{2+}$ only, indicating that $Mn^{2+}$ binding leads to a more homogenously folded P1.1 that inhibits the fast recruitment of NusA to PEC-104. The still two dissociation rate constants increased by ~2-fold upon addition of ligand, suggesting a competition between $Mn^{2+}$ and NusA binding. Overall, these observations demonstrate that ligand binding affects the early step of NusA-mediated enhancement of pausing by preventing initial recruitment of NusA to PEC-104. We were able to link these observations again to P1.1 switch helix formation, since the effect of $Mn^{2+}$ on NusA binding was lost for the MP1.1 mutant and largely restored for the compensatory MP1.1cp mutant (Fig. 5i, Supplementary Fig. 15 and Supplementary Table 5).

To ask whether the observed effect of ligand binding on NusA recruitment occurs during co-transcriptional folding, we next surveyed NusA binding during real-time transcription of the riboswitch[23,59] (Supplementary Fig. 16). In $Mg^{2+}$ only, and absence of $Mn^{2+}$, NusA remains bound to the active transcription complex for ~1.8 s, similarly to our results on the stalled PEC-104 under the same conditions (Supplementary Table 5). In contrast, when $Mn^{2+}$ is added co-transcriptionally, a slight, but significant decrease of the bound time is observed (~1.2 s; Supplementary Fig. 16c), indicating that co-transcriptional folding of the riboswitch in the presence of ligand alters NusA occupancy to the transcription complex in situ.

Collectively, our single-molecule and biochemical assays demonstrate that NusA-mediated enhancement of transcriptional pausing is abolished by $Mn^{2+}$-induced nucleation of P1.1, which in turn suppresses NusA recruitment to the PEC. That is, while functionally redundant in extending transcriptional pausing by favoring a stable P1.1, $Mn^{2+}$ and NusA do so in mutually exclusive fashion.

## NusA stabilizes the transient semi-docked riboswitch conformation

Our single-molecule NusA binding assay unveiled that ligand binding to the riboswitch prevents fast NusA recruitment to the PEC through P1.1 stabilization. While NusA binds transiently, as one of the most abundant transcription factors in the bacterial cell (at micromolar concentrations)[60] it is expected to be almost continuously bound to an active elongation complex (EC) in vivo[61] so that it is unlikely that PEC-104 exists in the absence of NusA. To understand the dynamics of the riboswitch in a context closer to in vivo conditions, we next evaluated the global docking dynamics of the NusA-bound PEC-104 using smFRET in the absence and presence of $Mn^{2+}$ (Fig. 6a).

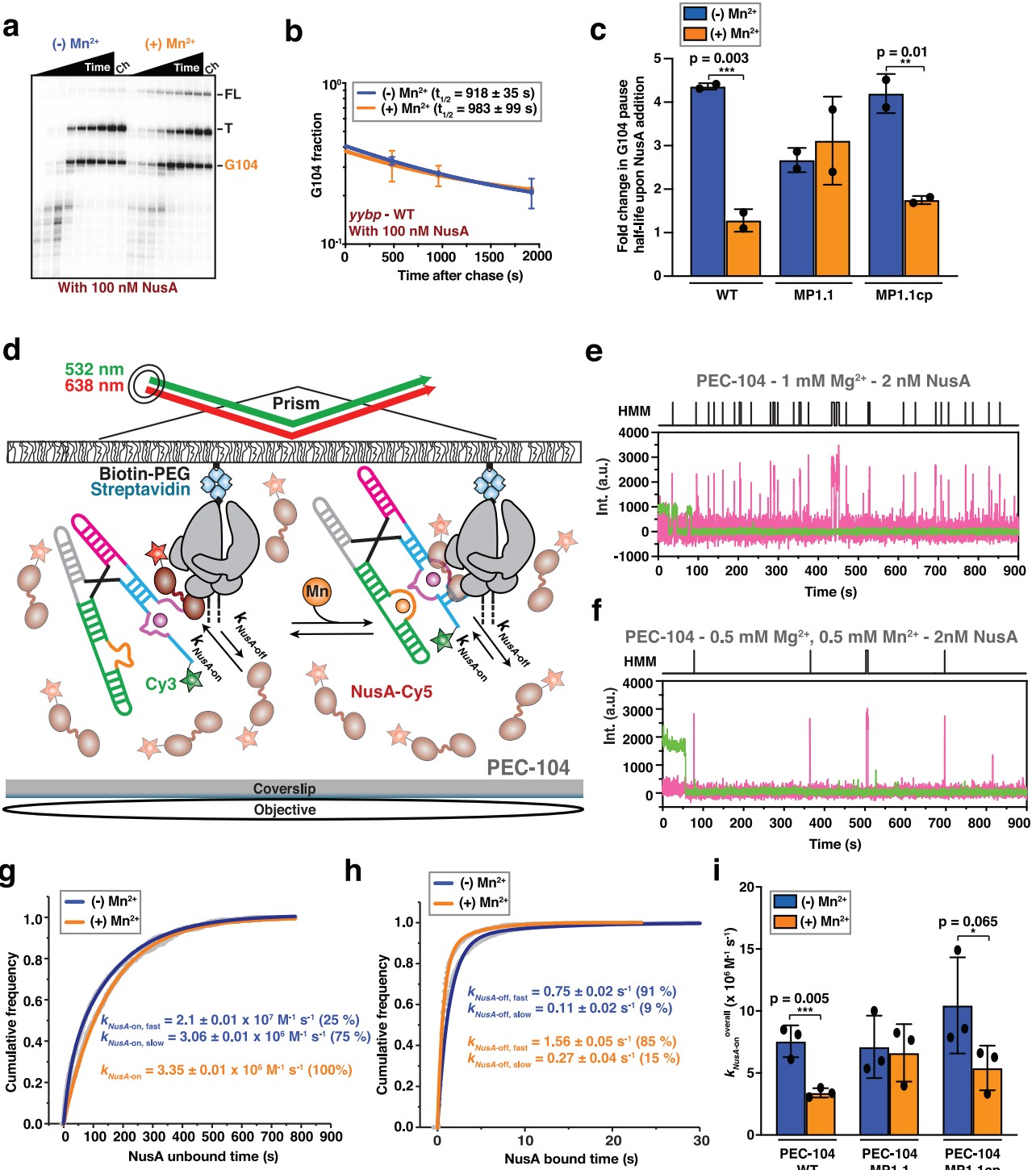

Similar to the traces collected for PEC-104 in the absence of $Mn^{2+}$, i.e., in the absence of any divalent ion or with $Mg^{2+}$ only, the NusA-saturated PEC-104 shows a high degree of dynamics between the three FRET states U, D* and D (Fig. 6b and Supplementary Fig. 17). In contrast, upon addition of ligand, the smFRET traces become more static, preferentially adopting the D* or D states (Fig. 6c), suggesting that both NusA and ligand stabilize the semi-docked and docked conformation of the riboswitch. Accordingly, in both the absence and presence of $Mn^{2+}$ with NusA bound the cumulative FRET histograms could be best fitted with three independent Gaussian peaks, corresponding to the three conformational states (U, D* and D; Fig. 6d, f). Moreover, upon $Mn^{2+}$ addition, the equilibrium shifted from the U

toward the D* state, rather than adopting the D state, further supporting that NusA particularly favors this intermediate semi-docked state.

Kinetic analysis of the transitions between the different conformational states further confirmed that the riboswitch was mostly dynamic in the absence of ligand. However, while the proportion of static molecules remained the same (29% in the absence versus 28% in the presence of NusA) with only $Mg^{2+}$, the transitions from U to either the D* or D states became much faster in the NusA-bound PEC compared to PEC-104 in the absence of the transcription factor (compare Fig. 6e and Fig. 4b and Supplementary Figs. 5 and 19). Indeed, both $k_{dock,U \to D}$ and $k_{dock,U \to D^*}$ significantly increased in the presence of

**Fig. 5 | Effect of Mn$^{2+}$ binding on NusA activity. a** Representative denaturing gel showing the RNAP pauses during the transcription of the Mn$^{2+}$ riboswitch in the presence of 100 nM NusA transcription factor. Position of the pause (G104), termination (T) and full-length (FL) products are indicated on the right. Experiments were performed using 25 μM rNTPs in the absence (−Mn$^{2+}$) and presence (+Mn$^{2+}$) of 0.5 mM Mn$^{2+}$. The chase lanes (Ch) were taken at the end of the time course after an additional incubation with 500 μM rNTPs for 5 min. Unprocessed images are provided in the Supplementary Information. **b** Fraction of complexes at the G104 pause in the presence of NusA transcription factor as a function of the transcription time in the absence (blue) and presence (orange) of 0.5 mM Mn$^{2+}$. The reported errors are the SD of the mean from $n = 2$ independent replicates. **c** Fold-change enhancement of G104 pause half-life in the WT, MP1.1 and MP1.1cp riboswitch variants obtained in the presence of 100 nM NusA factor relative to the same condition in the absence of the transcription factor. Experiments were performed using 25 μM rNTPs in the absence (blue) and presence (orange) of 0.5 mM Mn$^{2+}$. Error bars are the SD of the mean from $n = 2$ independent replicates. The statistical significance of differences was determined using the two-tailed Student's $t$-test (***$p < 0.01$, **$p < 0.05$, *$p < 0.1$). **d** Single-molecule setup for studying colocalization of NusA factor labeled with a single Cy5 fluorescent dye (NusA-Cy5) with a PEC. Immobilization of the PEC on the microscope slide occurs through the biotin-

streptavidin roadblock. Repeated bindings of NusA-Cy5 are monitored through direct excitation of the Cy5 fluorescent dye. Representative single-molecule trajectories showing the binding of NusA-Cy5 (pink) to PEC-104 recorded in the absence (**e**) or presence (**f**) of 0.5 mM Mn$^{2+}$ added co-transcriptionally. Hidden Markov modeling (HMM) is indicated on the top of each trace. A.U. arbitrary unit. Plots displaying the cumulative unbound (**g**) and bound (**h**) dwell times of NusA-Cy5 in the absence (blue) and presence (orange) of 0.5 mM Mn$^{2+}$ in the context of PEC-104. The binding ($k_{on}$) and dissociation ($k_{off}$) rate constants of NusA are indicated. The reported errors are the error of the fit. Total number of molecules analyzed for each condition is as follow: (−) Mn$^{2+}$ = 302; (+) Mn$^{2+}$ = 294. **i** Overall binding rate constants ($k_{on}$) of NusA-Cy5 in the context of PEC-104 WT, MP1.1 and MP1.1cp constructs determined in the absence (blue) and presence (orange) of 0.5 mM Mn$^{2+}$. Error bars are the SD of the mean from $n = 2$ independent replicates. The statistical significance of differences was determined using the two-tailed Student's $t$-test (***$p < 0.01$, **$p < 0.05$, *$p < 0.1$). See also Supplementary Fig. 13. Total number of molecules analyzed for each condition is as follow: PEC-104 WT (−) Mn$^{2+}$ = 302; PEC-104 WT (+) Mn$^{2+}$ = 294; PEC-104 MP1.1 (−) Mn$^{2+}$ = 232; PEC-104 MP1.1 (+) Mn$^{2+}$ = 303; PEC-104 MP1.1cp (−) Mn$^{2+}$ = 276; PEC-104 MP1.1cp (+) Mn$^{2+}$ = 301. Source data are provided with this paper at https://doi.org/10.7302/22513.

saturating transcription factor, suggesting that NusA promotes fast dynamics into both states (Fig. 6e). Further, the transitions between the D* and D states were observed to be the most probable (64%) with a faster regime toward the D state (5.63 s$^{-1}$; fraction 34% in the presence of NusA versus 3.03 s$^{-1}$; fraction 17% in the absence of NusA).

Upon addition of Mn$^{2+}$, the transition kinetics between the conformational states revealed that a 25% static population was found in SD* and 30% in SD (Fig. 6g), suggesting that NusA assists Mn$^{2+}$-induced stable docking. Most of the dynamic transitions (~64%) were observed between the D* and D states, with faster transitions toward the D state compared to PEC-104 without NusA-bound (4.66 s$^{-1}$; fraction 36% in the presence of NusA versus 2.44 s$^{-1}$; fraction 18% in the absence of NusA; Fig. 6g and Supplementary Fig. 20). Conversely, the transitions from the D to the D* state were significantly slower in the presence of both Mn$^{2+}$ and NusA (1.03 s$^{-1}$; fraction 28% in the presence of NusA versus 1.51 s$^{-1}$; fraction 16% in the absence of the transcription factor), supporting that NusA promotes full docking of the riboswitch through stabilization of the P1.1 switch helix. In further support, SiM-KARTS probing of P1.1 performed on NusA-bound PEC-104 revealed that this region is less accessible in the presence of the transcription factor (Supplementary Fig. 21 and Supplementary Table 3), confirming its impact factor on the local stabilization of P1.1.

### NusA preferentially recognizes the single-stranded P1.1 in the ligand-free conformation

Our single-molecule binding assays and structural probing so far unveiled a previously unappreciated effect of NusA transcription factor on nascent RNA structural dynamics and vice versa (Figs. 5 and 6). While the P1.1 switching helix is central for the overall gene regulation mediated by ligand binding, from transcriptional pausing to transcription termination, it is still unclear which conformational state is preferred for the initial recruitment of NusA. To probe the mechanistic details of NusA-mediated regulation of gene expression as a function of RNA structure, we next evaluated the kinetics of NusA binding to an "open" P1.1 helix using a 3-color microscopy experiment. Here, the open (single-stranded) state of P1.1 was surveyed through Cy3-labeled SiM-KARTS probe binding (green), while simultaneously observing the binding of Cy5-labeled NusA (pink) to PEC-104 (blue, Fig. 7a, b).

In the presence of only Mg$^{2+}$, NusA colocalized with the undocked P1.1 helix for ~1.4 s, a value very similar to the overall bound time of NusA to PEC-104 under the same conditions (Supplementary Table 5), supporting the notion that NusA captures the open P1.1 during its recruitment. Strikingly, upon addition of Mn$^{2+}$, this colocalization dwell time significantly decreased by ~2.5 fold (Fig. 7c), suggesting that folding of the P1.1 helix in the presence of ligand and NusA occupancy

are mutually exclusive. Control experiments surveying colocalization dwell times in the MP1 mutant of PEC-104 further corroborated that the effects on NusA association kinetics observed upon ligand binding require the ability to form the top base pairs of P1.1 (Fig. 7d).

Taken together, we discovered a previously unappreciated effect of NusA transcription factor on RNA structural dynamics, wherein binding of NusA to PEC-104 stabilizes an intermediate, semi-docked state of the riboswitch to assist downstream folding in the ligand-bound conformation. All of these effects involve the P1.1 switch helix as a focal point on which Mn$^{2+}$, NusA and RNAP all act; in return, P1.1 integrates these disparate signals as it modulates gene expression.

## Discussion

In this work, we have mechanistically dissected the co-transcriptional folding of the *L. lactis* Mn$^{2+}$ riboswitch, representative of the widespread *yybp* family, using a combination of transcription and single-molecule probing assays.

Taken together, our observations lead to the model in Fig. 8, wherein the signals of ligand binding and the ensuing tertiary structure docking of the aptamer domain, pausing by RNAP on a specific downstream DNA template sequence, and NusA transcription factor binding to the paused elongation complex are integrated by the central P1.1 switch helix to affect gene regulation. In this way, transcription-associated processes help fine-tune in real-time a local-to-global-to-local hierarchy of conformational dynamics of the nascent riboswitch during adaptive transcription. Within this molecular machine, the top two base pairs of the P1.1 helix fold in the RNAP exit channel and act as a central, nanoscale fulcrum, analogous to a first-class lever system, to finely balance all incoming signals for adaptive partitioning into either termination (without Mn$^{2+}$) or antitermination (with Mn$^{2+}$; Fig. 8). This molecular fulcrum mechanism empowers a bacterium to rapidly respond to an environmental threat by producing an exporter of excess Mn$^{2+}$[16]. Our mechanistic probing thereby significantly advances our understanding of the complex, tightly interwoven regulatory interaction network inside the bacterial cell that is all sensed and processed by the top base pairs of the P1.1 switch helix.

RNAP pausing has been known for its potential to facilitate co-transcriptional riboswitch folding, allowing time for the transcript to adopt a specific and functional conformation[7,21,22,62], and/or to bind its cognate ligand[17,51]. Transcriptional pausing classically has been understood to be stabilized by two main mechanisms: hairpin-stabilized (class I) and backtracked (class II)[36]. Class I pauses are further stabilized by transcription factor NusA upon interaction with the nascent RNA paused hairpin and the β-flap domain of the RNAP, resulting in the stabilization of the swiveled conformation of the

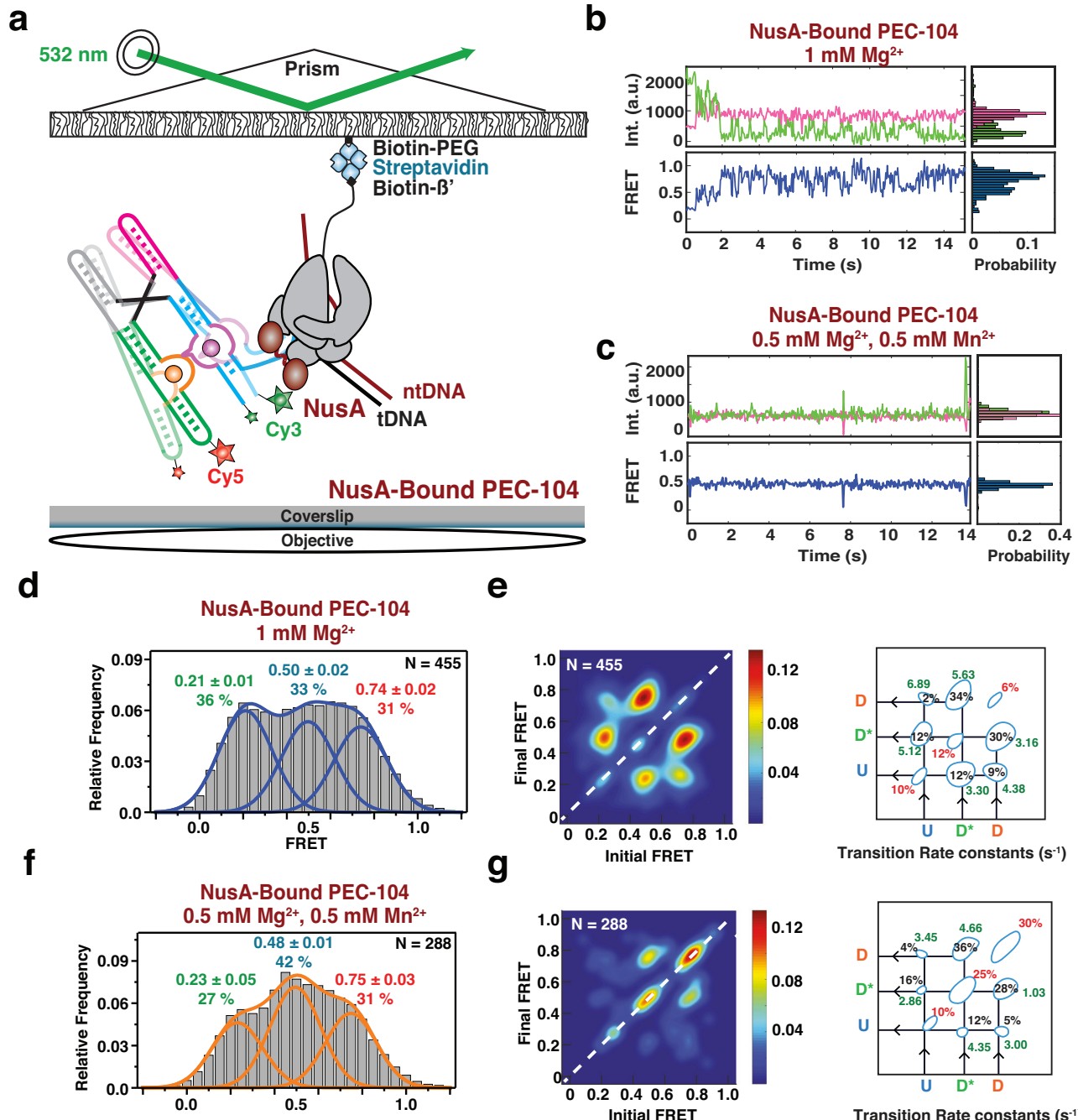

**Fig. 6 | Effect of NusA on the global conformation of the riboswitch. a** smFRET experimental setup. Paused Elongation Complex (PEC) immobilization utilizes biotinylated *E. coli* RNAP. Location of donor (Cy3 – green) and acceptor (Cy5 – red) fluorophores are indicated. Representative donor-acceptor (top) and smFRET (bottom) traces for NusA-Bound PEC-104 in the absence (**b**) and presence (**c**) of 0.5 mM $Mn^{2+}$. Probability FRET histogram for each trace is indicated on the right. SmFRET histograms of NusA-bound PEC-104 in the presence of only $Mg^{2+}$ (**d**) along with the corresponding Transition Occupancy Density Plots (TODP) and the determined transition rate constants (**e**). SmFRET histograms of NusA-bound PEC-104 in the presence of both $Mg^{2+}$ and $Mn^{2+}$ (**f**) along with the corresponding Transition Occupancy Density Plots (TODP) and the determined transition rate constants (**g**). The SD of each FRET value is reported. The total number of traces (N) included in each histogram is indicated on top right corner. See also Supplementary Figs. 16 and 17. TODPs represent dynamic traces as "off-diagonal" and the static traces as "on-diagonal" contour, where the color scale shows the prevalence of each population. The riboswitch conformational state corresponding to each FRET value is indicated. U = Undocked, D* = Pre-Docked, D = Docked. Source data are provided with this paper at https://doi.org/10.7302/22513.

RNAP[44,63]. Class II pauses instead are due to a reverse motion of the RNA relative to the enzyme (backtracking) and lead to the disengagement of the RNA 3' end from the RNAP catalytic site. This class of pause is sensitive to GreB and NusG, which promote RNA cleavage and suppress backtracking, respectively, for subsequent transcription reactivation[64,65]. More recently, a third class of transcriptional pausing (class III) has been identified in the $preQ_1$-sensing riboswitch from *B. subtilis*, where folding of an RNA pseudoknot structure inside the RNA exit channel modulates RNAP pausing efficiency distinctly in the absence and presence of ligand[21,25]. Our results here demonstrate that the $Mn^{2+}$ riboswitch G104 pause shares some similarities with the canonical class I pause, in that the nucleation of the top two base pairs

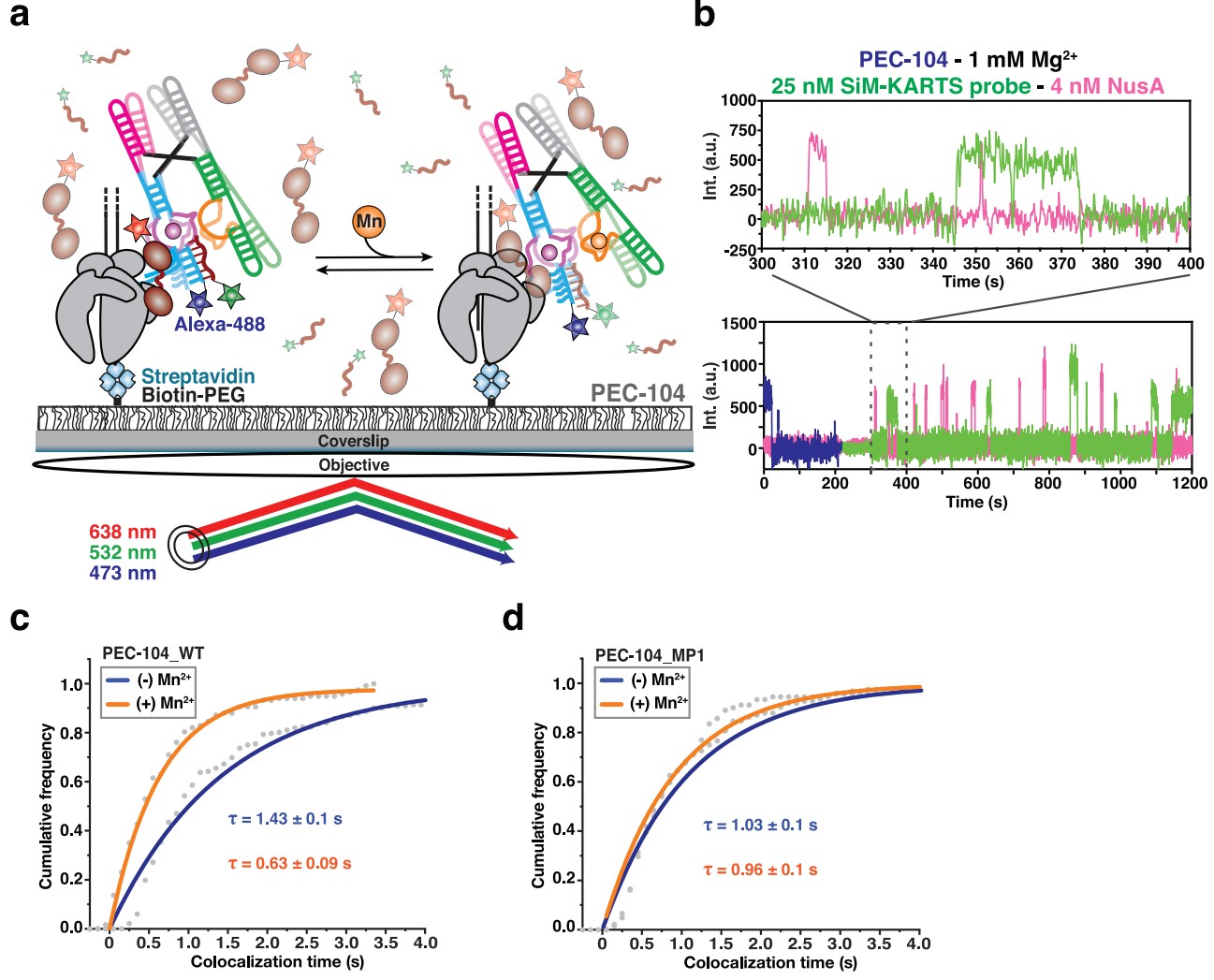

**Fig. 7 | NusA preferentially recognizes the open P1.1 helix. a** Single-molecule setup for studying colocalization of NusA factor labeled with a single Cy5 fluorescent dye (NusA-Cy5) and SiM-KARTS probe labeled with a single Cy3 fluorescent dye with a PEC. Immobilization of the PEC on the microscope slide occurs through the biotin-streptavidin roadblock. Repeated bindings of NusA and SiM-KARTS probe are monitored through direct excitation of the Cy3 and Cy5 fluorescent dyes. **b** Representative single-molecule trajectories showing the binding of NusA-Cy5 (pink) and SiM-KARTS probe (green) to single PEC-104 (blue). A.U., arbitrary unit. Plots displaying the cumulative colocalization time NusA-Cy5 and the SiM-KARTS probe in the absence (blue) and presence (orange) of 0.5 mM Mn$^{2+}$ in the context of PEC-104_WT (**c**) and PEC-104_MP1 (**d**). The colocalization dwell times are indicated. The reported errors are the errors calculated from bootstrapping of all dwell times collected. Total number of colocalization events analyzed for each condition is as follow: PEC-104_WT (−) Mn$^{2+}$ = 232; PEC-104_WT (+) Mn$^{2+}$ = 117; PEC-104_MP1 (−) Mn$^{2+}$ = 124; PEC-104_MP1 (+) Mn$^{2+}$ = 91. Source data are provided with this paper at https://doi.org/10.7302/22513.

of P1.1 stabilizes RNAP pausing. Notably, these base pairs reside at the edge of the RNA-DNA hybrid, within the RNA exit channel, in a configuration similar to the canonical class I *his* pause hairpin[66]. We also found that the G104 pause is enhanced in the presence of Mn$^{2+}$ ligand (Fig. 1), in parallel to the ligand-mediated stabilization of the P1.1 helix, analogous to the pausing mechanism found in the fluoride-sensing riboswitch[23]. In contrast to the class I pause, however, NusA is ejected from PEC-104 upon ligand binding and P1.1 stabilization (Fig. 5), perhaps because of a sterical clash with the docked remainder of the aptamer (Fig. 8). The first two G:C base pairs nucleate formation of the complete P1.1 switch helix and suppress the downstream strand invasion needed to form the overlapping terminator hairpin, reminiscent of the strand invasion mechanisms identified for other transcriptional riboswitches[67–69] (Fig. 8). These observations support a key role for transcription pausing exactly at G104 in setting the stage for adaptable P1.1 folding to regulate downstream gene expression. In this transcriptional context, pausing at G104 would allow more time for the riboswitch to sense its cognate ligand for accurate regulation of gene expression in response to environmental cues.

The structure of the *L. lactis* Mn$^{2+}$ riboswitch has been solved previously by X-ray crystallography[12]. In addition, the structure and dynamics of the closely related *Xanthomonas oryzae* Mn$^{2+}$ riboswitch were determined, highlighting the importance of adjacent Mg$^{2+}$ and Mn$^{2+}$ binding sites for tertiary structure docking of this elemental ion-sensing riboswitch in isolated form[13]. We found that the proximity of RNAP during co-transcriptional RNA folding, known to affect a range of RNA structure-based processes[70–72], is also a critical contributor to the ability of the nascent Mn$^{2+}$ riboswitch to regulate the outcome of gene expression, by stabilizing both global tertiary structure docking and local secondary structure (P1.1 helix) formation. Bacterial RNAP paused on its DNA template previously also has been shown to influence the adoption of transient transcriptional intermediates, allowing for more controlled sequential folding toward the regulatory structure[7,22,73]. Similarly, using smFRET we were able to determine that the intermediary, only semi-docked riboswitch state D* becomes prevalent in the PEC at the G104 pause site, perhaps indicating a direct steric clash with the proximal RNAP surface (Fig. 8). These findings are reminiscent of the folding pathway of the fluoride-sensing riboswitch

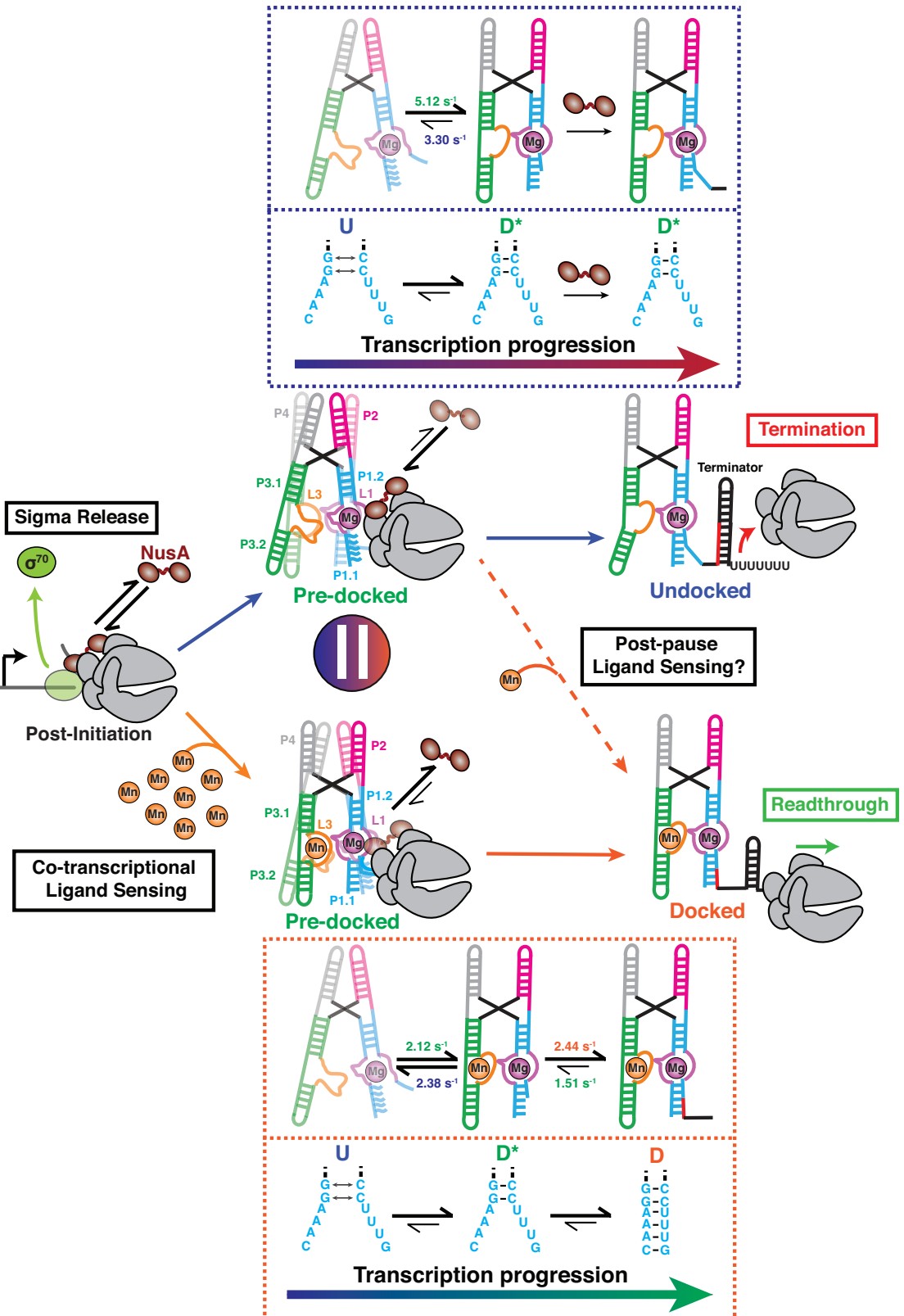

**Fig. 8 | Model of the co-transcriptional folding pathways of the *yybp* riboswitch class.** Transcription can follow two different pathways depending on the presence of a saturating concentration of Mn$^{2+}$ in the cellular environment. The G104 pause constitutes a transcriptional checkpoint at which NusA is either stabilized or released from the EC. In the absence of Mn$^{2+}$ (Upper), NusA remains bound to the EC assisting the initial nucleation of P1.1 switching helix. The pre-organized P1.1 helix remains in place to sense Mn$^{2+}$ at the latest stage of riboswitch transcription, allowing a fast response to environmental changes and efficiently regulate gene expression. Co-transcriptional folding of the riboswitch in the ligand-bound state (Lower), by contrast, promotes NusA release from the EC. Mn$^{2+}$ binding favors downstream stabilization of P1.1 to ultimately promote transcription readthrough.

in a similar paused complex[20], supporting a broad influence of RNAP on riboswitch folding. Such metastable conformational states additionally have been identified in other riboswitches, even in isolated form without RNAP[62,74,75], suggesting that pre-organization is a widespread strategy deployed by riboswitches for efficient ligand binding and folding in service of decisive regulation of gene expression.

Moreover, we found that the short-lived D* state is further stabilized by NusA binding to PEC-104, especially in the presence of ligand (Fig. 6). From the broader perspective of RNA folding in vivo, this observation highlights a versatile role of transcription factor NusA, proposed to assist the nucleation of RNA duplexes within the RNA exit channel[44], beyond the stabilization of imperfect intrinsic terminator hairpins[42,58]. Specifically, for the $Mn^{2+}$ riboswitch it performs a regulatory function upstream of the transcriptional decision point, similar to its role for the elemental fluoride ion-sensing riboswitch[23]. In addition, the D* state is undetectable once the entire P1.1 sequence is available upon release from or downstream transcription by the RNAP (Figs. 3 and 4). Taken together, these observations support a sensitive transcription regulation checkpoint at the G104 pause where ligand binding and the transcription machinery act in concert through fulcrum helix P1.1 to facilitate a finely balanced partitioning of the downstream expression platform into transcription termination or readthrough, depending on the input from multiple, integrated signals (Fig. 8).

The mechanistic single-molecule approaches developed here will undoubtedly help dissect many more intricate dynamic interplays between RNA folding and transcription, anticipated to govern the function of numerous non-coding RNAs across bacteria[5], with the potential to be leveraged for the design of novel anti-bacterial drugs[76].

## Methods
### DNA templates
A 249-nucleotide DNA template including the *yybp* riboswitch from *L. lactis* under the control of the T7A1 promoter was cloned into pUC19 plasmid between EcoRI and BamHI restriction sites. Transcription templates for in vitro transcription were generated by PCR using the "T7A1-PCR" forward oligonucleotide and the according reverse oligonucleotides. For stepwise transcription, the DNA template was generated by PCR using the "T7A1-Yybp-Start_AC" forward oligonucleotide and the "Yybp-EC104-DesthioBiotin" reverse oligonucleotide. For mutant DNA templates, the mutation was inserted in two PCR steps using overlapping oligonucleotides containing the corresponding mutation. MP1.1 and MP1.1cp show only transcription readthrough, independent of whether $Mn^{2+}$ is present or not, because the mutated 3' segment of P1.1 is also part of the terminator stem. Therefore, for the transcriptions presented in Supplementary Fig. 14c, the DNA templates additionally contained a corresponding mutation in the terminator stem, restoring terminator base pairing in the mutant contexts. The terminator stem mutations were introduced using oligonucleotides "Yybp-G122C/G123C (2)" and "Yybp-G122C/G123C (3)". Oligonucleotides used in this study are listed in Supplementary Table 1.

### In vitro transcription assays
Halted complexes (EC-13) were prepared in transcription buffer (20 mM Tris-HCl, pH 8.0, 20 mM NaCl, 1 mM $MgCl_2$, 14 mM 2-mercaptoethanol, 0.1 mM EDTA) containing 25 µM ATP/CTP mix, 50 nM $\alpha^{32}P$-GTP (3000 Ci/mmol), 10 µM ApU dinucleotide primer (Trilink), and 50 nM DNA template. *E. coli* RNAP holoenzyme (New England Biolabs) was added to 100 nM, and the mixture was incubated for 10 min at 37 °C. The sample was passed through G50 column to remove any free nucleotides. To complete the transcription reaction all four rNTPs were added concomitantly with heparin (450 µg/mL) to prevent the re-initiation of transcription. Time pausing experiments were performed using 25 µM rNTPs with 0.5 mM $MgCl_2$ (no ligand condition) or with 0.5 mM $MnCl_2$ (with ligand condition). The mixture was incubated at 37 °C, and reaction aliquots were quenched at the desired times into an equal volume of loading buffer (95% formamide, 1 mM EDTA, 0.1% SDS, 0.2% bromophenol blue, 0.2% xylene cyanol). Sequencing ladders were prepared by combining the halted complex with a chase solution containing 250 µM of each rNTP, in addition to one 3'-OMe rNTP (at 25 µM for 3'-OMe GTP and 15 µM for 3'-OMe ATP, UTP and CTP). Reaction aliquots were denatured before loading 5 µL of each onto a denaturing 8 M urea, 6% polyacrylamide sequencing gel. The gel was dried and exposed to a phosphor screen (typically overnight), which was then scanned on a Typhoon Phosphor Imager (GE Healthcare).

### Transcription data analysis
To determine the $T_{50}$ for regulation of termination by $Mn^{2+}$, the relative intensity of the full-length product was divided by the total amount of RNA transcript (full-length + terminated product) for each $Mn^{2+}$ concentration. The resulted percentages of readthrough were plotted against the $Mn^{2+}$ concentration using equation: $Y = Y_0 + ((M1 \times X)/(T_{50} + X))$, where $Y_0$ is the value of Y at $X_{min}$ (no ligand condition) and $M1 = Y_{max} - Y_{min}$.

The half-life of transcriptional pausing was determined by calculating the fraction of each RNA pause species compared with the total amount of RNA for each time point, which was analyzed with pseudo-first-order kinetics to extract the half-life[77]. For each determination, we have subtracted the background signal. Error bars in transcription quantification represent the standard deviation of the mean from independent replicates.

### NusA expression and purification
Wild-type and single-cysteine NusA were expressed from plasmids pNG5 and pKH3 respectively in BLR (DE3) cells. The cells were grown in LB media supplemented with 50 µg/mL kanamycin, induced with 1 mM IPTG when $OD_{600}$ reached ~0.7 and harvested 3 hours post-induction. His-Tagged proteins were purified using nickel affinity chromatography as described previously[45]. The fractions containing NusA protein were confirmed by SDS-PAGE and dialyzed into low salt dialysis buffer (20 mM Tris-HCl, pH 8.0, 200 mM NaCl, 0.5 mM TCEP, 0.1 mM $Na_2EDTA$) and subsequently purified using ion-exchange chromatography (Mono Q) using a NaCl gradient from 0.2 M to 2 M. In the case of the single-cysteine mutant, the ion-exchange chromatography was performed after the labeling step (see below). After purification, the protein was found to be purified to >95% homogeneity and mixed in 1:1 ratio with storage buffer (20 mM Tris-HCl, pH 8.0, 200 mM NaCl, 0.5 mM TCEP, 0.1 mM $Na_2EDTA$, 50% glycerol) and flash frozen in liquid nitrogen for storage at −80 °C.

### NusA labeling
NusA was labeled with Cy5-maleimide dye (PA25031, GE Healthcare) using an 8:1 molar ratio of dye to protein in a total volume of 200 µL (~20 µM protein) of labeling buffer (20 mM Tris-HCl, pH 8.0, 200 mM NaCl, 0.5 mM TCEP, 0.1 mM $Na_2EDTA$). After ~4.5 h incubation at 4 °C, the reaction was quenched upon addition of excess 2-mercaptoethanol. NusA-Cy5 and the free dye were separated by ion exchange chromatography. The labeling reaction was loaded on Mono Q column and washed with excess amount (over 20 column volume) of ion exchange buffer (20 mM Tris-HCl, pH 8.0, 200 mM NaCl, 0.5 mM TCEP, 0.1 mM Na2EDTA) to remove the free dye. Then NusA-Cy5 was eluted from the column using a NaCl gradient from 0.2 M to 2 M. The fractions containing NusA protein were confirmed by SDS-PAGE, the purified labeled protein was mixed in 1:1 ratio with storage buffer (20 mM Tris-HCl, pH 8.0, 200 mM NaCl, 0.5 mM TCEP, 0.1 mM $Na_2EDTA$, 50% glycerol) and flash frozen for storage at −80 °C. Labeling stoichiometry was determined as ~0.6 for Cy5/NusA using the absorbance measurements at $A_{280}$ and $A_{650}$.

## Preparation of fluorescently labeled nascent transcripts

In vitro transcription reactions were performed in two steps to allow the specific incorporation of Cy3 at the 5-end of the RNA sequence. Transcription reactions were performed in the same transcription buffer described above. Transcription reactions were initiated by adding 100 µM 5'-Cy3-ApU dinucleotide or 5'-ALEXA488-ApU for 3-color experiments and 25 µM rATP/rCTP/rGTP nucleotides at 37 °C for 10 min, thus yielding a fluorescent halted complex. The sample was next passed through a G50 column to remove any free nucleotides, and the transcription was resumed upon addition of all four rNTPs at the indicated concentration and heparin (450 µg/mL) to prevent the re-initiation of transcription. The resulted nascent transcript was hybridized to the 5' biotinylated CP (Anchor Bio oligonucleotide) complementary to the 3' end capture sequence, allowing immobilization of the complex to the microscope slide. The CP was mixed to a ratio of 10:1 with RNA transcript and added for 5 min before flowing the whole complex to the microscope slide. In the case of PEC transcription, the DNA templates contain a biotin at the 5' end of the template DNA strand. Streptavidin was mixed to a ratio of 5:1 with the DNA template for 5 min prior to start the transcription reaction. $Mn^{2+}$ (when present) was always added co-transcriptionally at a final concentration of 0.5 mM, and the same concentration of ligand was added into the corresponding buffer during subsequent dilutions.

## Preparation of fluorescently labeled PEC for smFRET

Two fluorescently labeled strands for the $Mn^{2+}$ sensing riboswitch from *L. lactis* were purchased from GE Darmacon. The two RNA strands were combined at a final concentration of 1 µM with either capture probe (LNA) or template DNA (tDNA) and non-template DNA (ntDNA) in nucleic acid scaffold buffer (50 mM Tris-HCl, pH 7.5, 100 mM KCl) at 1:1:1:1 ratio (RNA1:RNA2:tDNA:ntDNA). The solution was heated at 90 °C for 2 min followed by incubation at 37 °C for 10 min and then at room temperature for 10 min before storing at 4 °C. This annealed complex was stored at −20 °C for future experiments.

For experiments in the absence of RNAP, the annealing mixture was diluted at a concentration of 15–50 pM RNA and flowed onto a slide that had previously been passivated with a 10:1 ratio of PEG to biotinylated PEG, then incubated for 10–15 min with a 0.2 mg/mL solution of streptavidin.

For experiments performed in the presence of RNAP, the annealed nucleic acid scaffold was diluted to 50 nM RNA in assembly buffer (50 mM Tris-HCl, pH 7.5, 100 mM KCl, 1 mM MgCl₂), and *E. coli* RNAP was added to 250 nM. The mixture was incubated at 37 °C for 15 min and stored at 4 °C for the duration of single-molecule experiments. 50 pM of the reconstituted PEC was added to the slide for single-molecule imaging.

## Stepwise transcription followed by Click labeling for smFRET experiments

DNA templates for transcription were produced by PCR using oligonucleotides containing the T7A1 promoter sequence (Supplementary Table 1). The formation of stalled ECs was ensured by using DNA templates containing a Desthio-biotin at the 5' end of the antisense strand. The procedure is described in Supplementary Fig. 12. DNA templates (500 nM) were incubated with 20 µL Streptavidin-coated magnetic beads (Thermofisher Dynabeads M-270) overnight at room temperature in transcription buffer (20 mM Tris HCl pH 8.0, 20 mM MgCl2, 20 mM NaCl, 14 mM 2-mercaptoethanol, and 0.1 mM EDTA). In step 1, transcription reactions were initiated by adding 2 µM RNAP holoenzyme, 100 µM Cy3-ApC dinucleotide, 50 µM rATP/rGTP/rUTP-azido nucleotides at 37 °C for 20 min. The sample was washed three times with ten volumes of transcription buffer to remove any free nucleotides. In Step 2, RNAP was walked to the next position by adding 50 µM rCTP/rGTP/rUTP at 37 °C for 5 min. The azide-carrying ECs were incubated with 500 µM DBCO-Cy5 (Jena Bioscience) in the

transcription buffer for 1 h at 37 °C. ECs were then washed five times to remove unreacted dyes. In step 3, elongation to the roadblock position was achieved by adding 1 mM rNTPs at 37 °C for 5 min. To elute the PECs from the beads 10 µM competing biotinylated oligonucleotide was added at 37 °C for 20 min. Recovered complexes from the supernatant were directly flowed on a PEG/PEG-biotin treated microscope slide preincubated with 0.2 mg/mL streptavidin.

## Single-molecule experiments

All single-molecule fluorescence microscopy experiments were performed using a prism-based TIRF microscope based on an Olympus IX-71 frame equipped with a 60X- water-immersion objective (Olympus UPlanApo, 1.2 NA) or using the Oxford Nanoimager (ONI) microscope in TIRF (total internal reflection fluorescence) mode for 3-color experiments. All SiM-KARTS and NusA colocalization movies were collected at 100 ms time resolution and sm-FRET movies were collected at 60 ms time resolution (except for the P1.1 vector which was collected at 100 ms time resolution) using an intensified charge-coupled device camera (Hamamatsu C13440-20CU scientific complementary metal-oxide semiconductor camera). PEG-passivated quartz slides with a microfluidic channel containing inlet and outlet ports for buffer exchange were assembled as described in previous works[54,78]. The surface of the microfluidic channel was coated with streptavidin (0.2 mg/mL) for 10–15 min prior to flowing the nascent transcripts hybridized to the CP or PEC for smFRET. In the case of nascent transcripts analysis, the fluorescent PECs were directly injected into the channel surface using the biotin-streptavidin roadblock for immobilization.

For SiM-KARTS experiments, the nascent RNA transcripts were diluted in SiM-KARTS buffer (80 mM HEPES-KOH, pH 7.5, 300 mM KCl). For NusA colocalization assays, the molecules were diluted in imaging buffer (40 mM Tris-HCl, pH 8.0, 330 mM KCl, 0.1 mM EDTA, 0.1 mM dithiothreitol, 1 mg/mL bovine serum albumin). NusA-Cy5 (2 nM) or the SiM-KARTS probe (12.5 nM) was injected into the channel in the corresponding imaging buffer along with an enzymatic oxygen scavenging system consisting of 44 mM glucose, 165 U/mL glucose oxidase from *Aspergillus niger*, 2170 U/mL catalase from *Corynebacterium glutamicum*, and 5 mM Trolox to extend the lifetime of the fluorophores and to prevent photo-blinking of the dyes. The raw movies were collected for 15 min with direct green (532 nm) and red (638 nm) laser excitation for SiM-KARTS and NusA colocalization assays and 3 min with direct green (532 nm) for smFRET.

## Data acquisition and analysis for SiM-KARTS and NusA colocalization assays

Locations of molecules and fluorophore intensity traces for each molecule were extracted from raw movie files using custom-built MATLAB codes. Traces were manually selected for further analysis using the following criteria: single-step photobleaching of Cy3, ≥2 spikes of Cy5 fluorescence of more than 2-fold the background intensity. Traces showing binding events were idealized using a two-states (bound and unbound) model using a segmental *k*-means algorithm in QuB[79]. From the idealized traces, dwell times of NusA or the SiM-KARTS probe in the bound ($\tau_{bound}$) and the unbound ($\tau_{unbound}$) states were obtained. Cumulative plots of bound and unbound dwell-time distributions were plotted and fitted in Origin lab with single-exponential or double-exponential functions to obtain the lifetimes in the bound and unbound states. The dissociation rate constants ($k_{off}$) were calculated as the inverse of the $\tau_{bound}$, whereas the association rate constants ($k_{on}$) were calculated by dividing the inverse of the $\tau_{unbound}$ by the concentration of SiM-KARTS probe used during the data collection.

## Data acquisition and analysis for smFRET assay

Locations of molecules and fluorophore intensity traces for each molecule were extracted from raw movie files using custom-built

MATLAB codes. Single-molecule traces were then visualized using custom Matlab code and only those with a minimum combined intensity (Cy3 + Cy5 intensity) of 300 A.U., showing single-step photobleaching of the dyes, a signal-to-noise ratio of >3, and longer than 6 s were selected for further analysis. Selected traces were then background-subtracted to correct for crosstalk and (minimal) bleedthrough. We calculated the FRET ratio as $I_A/(I_A + I_D)$, where $I_A$ and $I_D$ are the background-corrected intensities of the acceptor (Cy5) and donor (Cy3), respectively. FRET histograms were made using the first 100 frames of all traces in each condition and fit with a sum of Gaussians using OriginPro 8.5. For kinetic analysis, traces were idealized with a three-state model corresponding to Undocked (low-FRET), Semi-Docked (mid-FRET) and Docked (high-FRET) states using the segmental k-means algorithm in QuB software as previously described[79]. Cumulative dwell-time histograms were plotted from all extracted dwell times and fit with single- or double-exponential functions using OriginPro 8.5 to obtain the lifetimes in the undocked ($\tau_{undock}$) and docked ($\tau_{dock}$) states. Rate constants of docking and undocking were then calculated as $k_{dock} = 1/\tau_{undock}$ and $k_{undock} = 1/\tau_{dock}$. For the double-exponential fits, kinetics were calculated similarly using both the short and long dwell lifetimes to obtain the fast and slow rate constants, respectively. The idealized smFRET traces were used for creating transition occupancy density plots (TODPs), which show the fraction of traces/molecules that exhibit a given type of transition at least once. In TODPs, dynamic traces showing a FRET transition (regardless of the number of transitions in that trace) and static traces (with no transitions over the entire trace) are weighted equally, avoiding overrepresentation of the traces with fast transitions.

## Reporting summary

Further information on research design is available in the Nature Portfolio Reporting Summary linked to this article.

## Data availability

All data supporting the findings of this study are available within the paper and its Supplementary Information. Uncropped gels are provided in the Supplementary Information. Source data are available through DeepBlue deposit: https://doi.org/10.7302/22513.

## Code availability

The code used in this study have been described in reference[79] and are available through DeepBlue deposit: https://doi.org/10.7302/22513.

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

## Acknowledgements

We thank Sebastian Velez for his precious help in optimizing the stepwise-click methodology, Alexey Kovalenko for the help in optimizing the three-color setup and Dr. Sujay Ray for helpful discussions. We thank Dr. Robert Landick (University of Wisconsin, Madison) for the generous gift of plasmids pNG5 and pKH3 for the purification of NusA wild-type and single-cysteine mutant. This work was supported by NIH grants R01 GM118524 and MIRA R35 GM131922, as well as NSF grant MCB 2140320, to N.G.W.

## Author contributions

Conceptualization, A.C. and N.G.W.; Methodology, A.C.; Investigation, A.C., S.D., R.R.; Writing - Original Draft, A.C.; Writing - Review & Editing, A.C., S.D., R.R., N.G.W.; Supervision, N.G.W; Funding Acquisition, N.G.W.

## Competing interests

The authors declare no competing interests.
