## [Peer Review File · Nature Communications]

A nascent riboswitch helix orchestrates robust transcriptional regulation through signal integrationReviewers' Comments:

Reviewer #1:

Remarks to the Author:

In this study, Chauvier et al. carry out creative single molecule and biochemical approaches to characterize the Mn²⁺ sensing riboswitch of *Lactococcus lactis*, examining the interplay between transcription, RNA folding and binding of the NusA termination factor and the riboswitch ligand manganese.

The authors carried out many careful experiments, however, the study comes to some of the same conclusions reported by the same group for a fluoride-sensing riboswitch (PMID: 34782462).

I have several suggestions for making the study more accessible to a broad audience.

--The results and discussion sections are very long and should be condensed. It is easy to get lost in all the experimental details and numbers such that the big picture concepts are missed, though the summary sentences at the end of each subsection are helpful.

--The authors should tone down hyperbole such as "a molecular fulcrum of a first-class lever system" (abstract). I don't even know what this means with respect to this study. Is the *L. lactis* riboswitch really "paradigmatic"?

--The authors extensively cite their own publications but overlook relevant papers from other groups such as PMID: 25794618, PMID: 29805037, PMID: 36150954.

--The authors should use color more judiciously in all their figures. In particular, the most important points are lost in the hyper-colored model panels (Fig. 2b, 3a, 5d, 6a and 7a). For example, there is no reason for RNA polymerase to be three different colors and the green background should be eliminated. The RNA structures and positions of the fluorophores should be most obvious in these panels.

--Page 12, Lines 236-237: "in Mg²⁺ only with RNAP removed" is confusing.

Reviewer #2:

Remarks to the Author:

This is a great study delving into the mechanism of folding of the yybP-ykoY RNA, a riboswitch enriched in the genomes of human and plant pathogens. Understanding the mechanism of this riboswitch is clearly important for consideration of novel therapeutics and the folding pattern is not well understood, until now. The authors emphasize an important physical property of many riboswitches, co-transcriptional folding in the presence of cognate ligands, the connection to transcription factors, transcription pauses, and RNAP occupancy.

Here are my comments:

Page 8, Line 151: With such a short antisense oligo used in the SiM-KARTS analysis, how do we know that there is specific binding to the P1.1 stem? Antisense oligos can nucleate other areas of the RNA. Were other antisense oligos, of perhaps longer lengths, used to test their effects on riboswitch binding?

Page 12, Line 238: Where is figure "3E"? I think the lettering stops at "D."

Page 13, Line 253: "Taken together, our analysis so far reveals that the presence of RNAP at the G104 pause site promotes the docked riboswitch conformation, particularly in the absence of ligand, due to dynamic sampling of a partially folded P1.1 switch helix." This is a great insight, highlighting the novelty of this paper.

Page 15, Line 303: Could the authors elaborate a little more on the mechanism by which they expect RNAP to affect folding of P1.1 directly? They cite some previous studies on the preQ1 switch but how would this switch work with RNAP?

Page 16, Line 323: The authors do not discuss the DBCO click reaction conditions in the methods section.

Page 28, Line 590: Change "249-nucleotides" to "249-nucleotide"

Page 28, Line 599: "MP1.1 and MP1.1cp shows only transcription readthrough regardless of the presence of Mn2+." Could you please rephrase? This is unclear

Page 29, Line 607: The use of 14mM Mercaptoethanol is unusual. Wouldn't this break apart the RNAP complex?

Reviewer #3:

Remarks to the Author:

Chauvier et al present a detailed investigation of how a riboswitch responds to Mn2+ during transcription. The study builds on previous work on this riboswitch in the Walter lab and makes expert use of the various single molecule fluorescence tools they have developed. At this point, there have been several papers on the conformational dynamics of riboswitches. The present study is significant for the following reasons: First, it is another example of how the folding dynamics of the nascent RNA in general, and the P1 switch helix in particular, are poised to respond quickly to ligand binding during transcription. This now adds to the examples of riboswitches in which strand exchange is intricately coupled to RNA elongation. Second, the authors show how an intrinsic transcription pause extends the window in which the P1 helix samples "on" and "off" conformations. Lastly, the authors show that interactions with RNAP itself, and with NusA elongation factor, also influence the conformation of the nascent RNA, favoring (or disfavoring) base pairing of P1.1 and thus the likelihood of transcription readthrough in response to Mn2+. The last point extends the current picture of co-transcriptional folding and termination mechanisms and is particularly interesting.

The data are high quality, and the analyses are very thorough. I don't have any major concerns. Minor suggestions for clarifying the model and the presentation are listed below.

1. The authors conclude that Mn2+ is mainly sensed before the transcription pause at G104 is reached, rather than afterwards. I didn't follow the logic for this assumption – is it known that Mn2+ can already complex with the P3 domain upstream? Otherwise, if the pre-docked intermediate ensemble is the same with or without Mn2+, it seems that sensing occurs DURING the pause, and this is the whole point of the intrinsic pause at this strategic position. (If the pause does not enable ligand sensing and anti-termination, it is not at all clear on line 124 why RNAP pausing upstream of the terminator / anti-terminator is helpful since P1.1 is not fully stable until RNAP advances past G104). It would be very helpful if the authors could clarify this key aspect of the model in the results and discussion and consider redrawing the scheme in Fig. 8 as needed.

2. Related to the point above, the paper begins by observing that Mn2+ stabilizes the paused transcription complex. Mn2+ inherently increases pausing of many polymerases. Did the authors verify that pause stabilization in Fig. 1 depends on Mn2+-induced folding of the aptamer domain? One way to do this might be to check transcription with and without Mn2+ of a P3 mutant that cannot bind Mn2+ or a mutant that cannot form the aptamer tertiary structure. The P1.1 mutations show that the P1.1 stem contributes to the intrinsic pause, but not whether stabilization of this pause arises from Mn2+ binding to the RNAP active site vs. the aptamer domain. (Even if stabilization of the pause requires Mn2+-dependent folding of the aptamer domain, the authors still can't know if Mn2+ binds before or after the pre-docked state forms.)

3. Line 258 – this paragraph is difficult to follow as written, rewrite for clarity. (For example, a statement about static docking is followed by a statement about dynamic docking / undocking; the relation between the two is not obvious.). It might be less confusing and more economical to immediately introduce the five conformational states detected by HMM modeling. As I understand it, the take home point of this section is that paused RNAP downstream of P1.1 shifts the energy landscape so that the RNA exchanges among intermediate conformations, poisoning the riboswitch to respond to Mn2+ (at least, this is what I see when I look at Figure 4.) This is an interesting observation but hard to grasp from the detailed and lengthy description of Figure 4.

4. The experiments in Fig. 6-7 showing how NusA stabilizes the open P1.1 whereas Mn²⁺ stabilizes the closed P1.1 is super exciting, but this point is buried in the very long and detailed description of the results in Fig. 5-7. For the benefit of a broader audience, it would be ideal if the authors could streamline their presentation of these results (without losing too much precision). I also recommend shortening the discussion (avoiding too much repetition of the results).

5. Line 159 – do single transcripts sample both conformations accessible to the SiM-KARTS probe, or are there two (non-exchanging) populations of RNA molecules? It would be helpful to clarify this at this point in the manuscript, so the reader is better prepared for the next parts of the story.

6. The conclusion that the semi-docked intermediate represents a partially paired P1.1 (from Fig. S13) is probably right but the FRET evidence for this is indirect as the fluorophores read out whether P1 is stable or not, but not how much of it has paired. It would be helpful to acknowledge this uncertainty in the assignment of this intermediate state (or ensemble of states).

7. Line 320, what does it mean to “access the P1.1 FRET vector”?
Line 336, “steric”, not “sterical”

Reviewer #1 (Remarks to the Author):

In this study, Chauvier et al. carry out creative single molecule and biochemical approaches to characterize the Mn²⁺ sensing riboswitch of *Lactococcus lactis*, examining the interplay between transcription, RNA folding and binding of the NusA termination factor and the riboswitch ligand manganese.

The authors carried out many careful experiments, however, the study comes to some of the same conclusions reported by the same group for a fluoride-sensing riboswitch (PMID: 34782462).

Response: We thank the reviewer for the constructive comments. We agree with their notion that some of the conclusions unveiled in this study are relatively similar to some of the ones reported for the fluoride-sensing riboswitch with respect to the impact of RNA folding on transcriptional pausing and transcription factor NusA recruitment to the Pause Elongation Complex (PMID: 34782462). This other example of regulation induced by co-transcriptional riboswitch folding and ligand binding to the aptamer domain in fact demonstrates the broader relevance of our current study. New in the present study is that we define a previously uncharacterized intermediate riboswitch fold that only could be observed in the context of a specific Paused Elongation Complex. In addition, we unveiled the impact of transcription factor NusA on riboswitch folding dynamics which, to our knowledge, has never been characterized before. We have made sure to emphasize the novel findings in the revised version of the manuscript.

I have several suggestions for making the study more accessible to a broad audience. --The results and discussion sections are very long and should be condensed. It is easy to get lost in all the experimental details and numbers such that the big picture concepts are missed, though the summary sentences at the end of each subsection are helpful.

Response: We thank the reviewer for this very helpful comment. As suggested, we have shortened the Results and Discussion sections to focus on the results and main concepts rather than describing experimental procedures in detail. For example, we removed the extensive description of the preparation of fluorescently labeled Paused Elongation Complexes and the SiM-KARTS methodology since it was used in a previous study (PMID: 34782462) and described in a *Methods* paper (PMID: 30951833).

--The authors should tone down hyperbole such as “a molecular fulcrum of a first-class lever system” (abstract). I don’t even know what this means with respect to this study. Is the *L. lactis* riboswitch really “paradigmatic”?

Response: We appreciate this assessment and have replaced the word “paradigmatic” with “representative” as the *L. lactis* riboswitch is a well-studied example of the large class of manganese riboswitches. Concerning the analogy of the “fulcrum”, we feel that it is useful for the reader to have a physical picture of the fine balance that the manganese riboswitch achieves between multiple inputs from ligand, protein cofactor and RNA polymerase. Since the other reviewers did not have a problem with the expression, we would respectfully prefer keeping it.

--The authors extensively cite their own publications but overlook relevant papers from other groups such as PMID: 25794618, PMID: 29805037, PMID: 36150954.

Response: We now have incorporated the suggested references in the text where appropriate and thank the reviewer for these very helpful additions.

--The authors should use color more judiciously in all their figures. In particular, the most important points are lost in the hyper-colored model panels (Fig. 2b, 3a, 5d, 6a and 7a). For example, there is no reason for RNA polymerase to be three different colors and the green background should be eliminated. The RNA structures and positions of the fluorophores should be most obvious in these panels.

Response: This point is well taken. We have now simplified the figures as suggested by the reviewer, removing the color from RNA polymerase and background.

--Page 12, Lines 236-237: "in Mg²⁺ only with RNAP removed" is confusing.

Response: To improve clarity, we replaced the term "removed" as follows: "Notably, in the presence of Mg²⁺ ions only, the docked population without RNAP was significantly less populated..."

Reviewer #2 (Remarks to the Author):

This is a great study delving into the mechanism of folding of the yybP-ykoY RNA, a riboswitch enriched in the genomes of human and plant pathogens. Understanding the mechanism of this riboswitch is clearly important for consideration of novel therapeutics and the folding pattern is not well understood, until now. The authors emphasize an important physical property of many riboswitches, co-transcriptional folding in the presence of cognate ligands, the connection to transcription factors, transcription pauses, and RNAP occupancy.

Response: We thank the reviewers for their kind and constructive comments.

Here are my comments:

Page 8, Line 151: With such a short antisense oligo used in the SiM-KARTS analysis, how do we know that there is specific binding to the P1.1 stem? Antisense oligos can nucleate other areas of the RNA. Were other antisense oligos, of perhaps longer lengths, used to test their effects on riboswitch binding?

Response: Single Molecule Kinetic Analysis of RNA Transient Structure (or SiM-KARTS) relies on the specific and transient interaction of a DNA probe in order to survey structural changes in the RNA transcript as a function of the imaging conditions (PMID: 26781350 and 30951033). In order to be specific to the probed RNA sequence (P1.1 stem), we used a relatively longer probe (8 nucleotides) compared to our initial study presenting the method (6 complementary nucleotides). However, increasing the probe

length further can result in binding of the probe to the target that is too stable. This would prevent the extraction of the valuable kinetic parameters and could potentially alter the structure of the interrogated riboswitch, as mentioned by the reviewer below. Because the kinetics of the SiM-KARTS probe binding changed when we introduced mutations (MP1) within the targeted RNA switch helix, we are confident that the probe is specific to this region of the riboswitch.

Page 12, Line 238: Where is figure “3E”? I think the lettering stops at “D.”

Response: We thank the reviewer for catching this typographical error within the figure in which the label “E” was missing. We have corrected it in the revised manuscript.

Page 13, Line 253: “Taken together, our analysis so far reveals that the presence of RNAP at the G104 pause site promotes the docked riboswitch conformation, particularly in the absence of ligand, due to dynamic sampling of a partially folded P1.1 switch helix.” This is a great insight, highlighting the novelty of this paper.

Response: We thank the reviewer for this kind praise.

Page 15, Line 303: Could the authors elaborate a little more on the mechanism by which they expect RNAP to affect folding of P1.1 directly? They cite some previous studies on the preQ1 switch but how would this switch work with RNAP?

Response: In previous studies, our group characterized the impact of RNAP pausing on riboswitch folding and vice versa using the preQ₁-sensing riboswitch as a model system (PMID: 30388413 & 37264140). In this earlier work, we found that the presence of RNAP at the pause position stabilizes the P2 stem due to RNA-protein interactions within the RNA exit channel. In the present study, we propose that the presence of RNAP at the G104 pause position promotes P1.1 stabilization in a similar way. However, because we currently lack a high-resolution structure of this specific Paused Elongation Complex, we cannot pinpoint those specific interactions and how they would affect riboswitch folding locally and generally. We made this statement clear in the revised manuscript, as follows: “[...] due to the absence of an atomic structure of the paused elongation complex studied here, we lack a detailed picture of the diverse interactions within the RNAP’s exit channel that involve both RNA transcript and NusA transcription factor. Structural determination of such complexes will be needed to pinpoint the interfaces involved in such an intriguing mechanism of pausing and unpausing.”

We kept the narrative in the results section to mention only that the presented results are suggesting electrostatic or steric interaction with RNAP: “This observation suggests that the RNAP can stabilize the semi-docked state, possibly through electrostatic and steric interaction with the riboswitch...”

Page 16, Line 323: The authors do not discuss the DBCO click reaction conditions in the methods section.

Response: We now have included a description of the DBCO-click reaction in the Methods section.

Page 28, Line 590: Change “249-nucleotides” to “249-nucleotide”

Response: We made this correction in the manuscript.

Page 28, Line 599: “MP1.1 and MP1.1cp shows only transcription readthrough regardless of the presence of Mn²⁺.” Could you please rephrase? This is unclear.

Response: We rephrased this sentence as follows: “MP1.1 and MP1.1cp show only transcription readthrough, independent of whether Mn²⁺ is present or not, because the mutated 3' segment of P1.1 is also part of the terminator stem. Therefore, for the transcriptions presented in Supplementary Figure 14c, the DNA templates additionally contained a corresponding mutation in the terminator stem, restoring terminator base pairing in the mutant contexts.”

Page 29, Line 607: The use of 14mM Mercaptoethanol is unusual. Wouldn't this break apart the RNAP complex?

Response: The buffer used for bulk *in-vitro* transcription has been deployed in the past in numerous studies (PMID: 22331895, 28071751, 33850018, 34782462). In addition, control experiments supporting that the Paused Elongation Complex is stable (> 48 hours) under these conditions have been performed previously (PMID: 28071751 & 31081713).

Reviewer #3 (Remarks to the Author):

Chauvier et al present a detailed investigation of how a riboswitch responds to Mn²⁺ during transcription. The study builds on previous work on this riboswitch in the Walter lab and makes expert use of the various single molecule fluorescence tools they have developed. At this point, there have been several papers on the conformational dynamics of riboswitches. The present study is significant for the following reasons: First, it is another example of how the folding dynamics of the nascent RNA in general, and the P1 switch helix in particular, are poised to respond quickly to ligand binding during transcription. This now adds to the examples of riboswitches in which strand exchange is intricately coupled to RNA elongation. Second, the authors show how an intrinsic transcription pause extends the window in which the P1 helix samples “on” and “off” conformations. Lastly, the authors show that interactions with RNAP itself, and with NusA elongation factor, also influence the conformation of the nascent RNA, favoring (or disfavoring) base pairing of P1.1 and thus the likelihood of transcription readthrough in response to Mn²⁺. The last point extends the current picture of co-transcriptional folding and termination mechanisms and is particularly interesting.

The data are high quality, and the analyses are very thorough. I don't have any major concerns. Minor suggestions for clarifying the model and the presentation are listed

below.

Response: We thank the reviewer for the kind comments and the constructive remarks to help improve the clarity of the manuscript. We performed the necessary changes as described below.

1. The authors conclude that Mn²⁺ is mainly sensed before the transcription pause at G104 is reached, rather than afterwards. I didn't follow the logic for this assumption – is it known that Mn²⁺ can already complex with the P3 domain upstream? Otherwise, if the pre-docked intermediate ensemble is the same with or without Mn²⁺, it seems that sensing occurs DURING the pause, and this is the whole point of the intrinsic pause at this strategic position. (If the pause does not enable ligand sensing and anti-termination, it is not at all clear on line 124 why RNAP pausing upstream of the terminator / anti-terminator is helpful since P1.1 is not fully stable until RNAP advances past G104). It would be very helpful if the authors could clarify this key aspect of the model in the results and discussion and consider redrawing the scheme in Fig. 8 as needed.

Response: The reviewer is right in that we have no indication that Mn²⁺ sensing can occur before the G104 pause is reached during the transcription process. Considering the idea that this transcriptional pause constitutes an example of a transcriptional checkpoint, we agree with the reviewer that it makes sense that the ligand sensing would occur during the pause. To avoid confusion, we added the following statement in the discussion section: “In this transcriptional context, pausing at G104 would allow more time for the riboswitch to sense its cognate ligand for accurate regulation of gene expression in response to environmental cues.”

2. Related to the point above, the paper begins by observing that Mn²⁺ stabilizes the paused transcription complex. Mn²⁺ inherently increases pausing of many polymerases. Did the authors verify that pause stabilization in Fig. 1 depends on Mn²⁺-induced folding of the aptamer domain? One way to do this might be to check transcription with and without Mn²⁺ of a P3 mutant that cannot bind Mn²⁺ or a mutant that cannot form the aptamer tertiary structure. The P1.1 mutations show that the P1.1 stem contributes to the intrinsic pause, but not whether stabilization of this pause arises from Mn²⁺ binding to the RNAP active site vs. the aptamer domain. (Even if stabilization of the pause requires Mn²⁺-dependent folding of the aptamer domain, the authors still can't know if Mn²⁺ binds before or after the pre-docked state forms.)

Response: We thank the reviewer for this constructive remark. We agree that Mn²⁺ ions have the potential to affect the transcription rate (and thereby transcriptional pausing) by interacting with the RNAP active site. In our study, we used sub-millimolar concentrations of Mn²⁺ ion to limit this potential complication. However, to further support that the enhancement of pausing efficiency is due to Mn²⁺ binding to the riboswitch, we performed the suggested control experiment. Specifically, we performed the time pausing assay using a mutant (A41U) of the loop-loop interaction previously identified as unresponsive to Mn²⁺ (PMID: 31541094). As seen in the presented representative gel below, this variant is unresponsive to the presence of Mn²⁺, both in

G104 pause behavior and transcription readthrough, supporting that the observed effect described in Figure 1 is due to P1.1 folding as a function of ligand binding to the riboswitch.

These results are now included in the revised version of the manuscript in the first paragraph of the Results as follows: “Control experiments performed with a mutant abolishing Mn²⁺ binding to the riboswitch (A41U)¹³ showed no such difference, supporting that promotion of RNAP pausing is the result of riboswitch folding in the ligand-bound state, rather than a direct effect on the enzyme (Supplementary Table 2).”

3. Line 258 – this paragraph is difficult to follow as written, rewrite for clarity. (For example, a statement about static docking is followed by a statement about dynamic docking / undocking; the relation between the two is not obvious.). It might be less confusing and more economical to immediately introduce the five conformational states detected by HMM modeling. As I understand it, the take home point of this section is that paused RNAP downstream of P1.1 shifts the energy landscape so that the RNA exchanges among intermediate conformations, poising the riboswitch to respond to Mn²⁺ (at least, this is what I see when I look at Figure 4.) This is an interesting observation but hard to grasp from the detailed and lengthy description of Figure 4.

Response: We thank the reviewer for this constructive remark. We rephrased the paragraph to make the initial findings clearer to a broad audience:

“From our previous work it is known that binding of Mn²⁺ stabilizes a static docked (SD) conformation in the riboswitch in the presence of a complete P1.1 stem¹³. Further inspection of the smFRET trajectories in our 104 constructs revealed that HMM modeling was best performed using three FRET states (Undocked = U (~0.2), Semi-Docked = D* (~0.45), Docked = D (~0.75)).”

4. The experiments in Fig. 6-7 showing how NusA stabilizes the open P1.1 whereas Mn²⁺ stabilizes the closed P1.1 is super exciting, but this point is buried in the very long and detailed description of the results in Fig. 5-7. For the benefit of a broader audience, it would be ideal if the authors could streamline their presentation of these results (without losing too much precision). I also recommend shortening the discussion (avoiding too much repetition of the results).

Response: We thank the reviewer for this constructive suggestion. We shortened the Results section to emphasize the findings rather than the experimental procedures, as also suggested by Reviewer #1. For example, we removed the extensive description of fluorescently labeled Paused Elongation Complexes and SiM-KARTS procedures. Now the results section starts as follows:

“To characterize the co-transcriptional folding of the riboswitch P1.1 helix as a function of ligand binding, we used an *in vitro* transcription assay that generates a co-transcriptionally folded PEC with a single fluorophore at the 5' end for identification and subsequent probing at the single molecule level (Figure 2a and b)²².”

In addition, we removed the summary of the results from the Discussion section to directly introduce and present the model as follow:

“In this work, we have mechanistically dissected the co-transcriptional folding of the *L. lactis* Mn²⁺ riboswitch, representative of the widespread *yybp* family, using a combination of transcription and single molecule probing assays.”

5. Line 159 – do single transcripts sample both conformations accessible to the SiM-KARTS probe, or are there two (non-exchanging) populations of RNA molecules? It would be helpful to clarify this at this point in the manuscript, so the reader is better prepared for the next parts of the story.

Response: The two scenarios mentioned by the reviewer are entirely possible. However, with the experiments performed and presented in this manuscript, we cannot rule out one case versus the other solely based on SiM-KARTS. We agree that this piece of information is rather unveiled by the single molecule FRET experiments. We rephrased our statements and conclusions to help the reader being prepared for this next part of the story:

“This observation is consistent with the notion that the SiM-KARTS probe senses (at least) two alternative structures of the riboswitch P1.1 helix that could arise on the same RNA molecule.”

“Our biochemical and single molecule probing assays underscore the importance of both RNAP and ligand for riboswitch folding. However, SiM-KARTS only reveals structural rearrangements that occur locally (P1.1 stem) and does not report the intramolecular dynamics for a single RNA molecule. Generally, the presence of RNAP and DNA template during transcription has been shown to affect global folding of structural RNAs.”

6. The conclusion that the semi-docked intermediate represents a partially paired P1.1 (from Fig. S13) is probably right but the FRET evidence for this is indirect as the fluorophores read out whether P1 is stable or not, but not how much of it has paired. It would be helpful to acknowledge this uncertainty in the assignment of this intermediate state (or ensemble of states).

Response: We agree with the reviewer that the smFRET evidence is indirect and limited by the observation of a single end-to-end fluorophore distance rather than

specific base pairing. We rephrase the presentation of the FRET vector to avoid this confusion:

To further test whether the observed conformational changes are due to stabilization of switching helix P1.1 in PEC-104, we directly surveyed P1.1 folding using smFRET.” We then kept the discussion of P1.1 stability as it was initially discussed.

7. Line 320, what does it mean to “access the P1.1 FRET vector”?

Line 336, “steric”, not “sterical”

Response: To improve the clarity of this statement, we revised the phrase “access the P1.1 FRET vector” as follows: “To measure dynamic distances along the P1.1 helical axis, we employed stepwise transcription...”

Reviewers' Comments:

Reviewer #3:

Remarks to the Author:

The authors have addressed all of my comments on the previous version, and added a helpful control. The text is still long but the figures are easier to read. This study advances the understanding of riboswitch mechanisms and also transcription pausing with or without NusA.